# Fibril-induced glutamine-/asparagine-rich prions recruit stress granule proteins in mammalian cells

Katrin Riemschoss[1], Verena Arndt[1], Benedetta Bolognesi[2] , Philipp von Eisenhart-Rothe[1], Shu Liu[1], Oleksandra Buravlova[1], Yvonne Duernberger[1], Lydia Paulsen[1], Annika Hornberger[1], André Hossinger[1], Nieves Lorenzo-Gotor[2] , Sebastian Hogl[3], Stephan A Müller[3,7], Gian Tartaglia[2,4,5], Stefan F Lichtenthaler[3,6,7], Ina M Vorberg[1,8]

Prions of lower eukaryotes are self-templating protein aggregates that replicate by converting homotypic proteins into stable, tightly packed beta-sheet–rich protein assemblies. Propagation is mediated by prion domains, low-complexity regions enriched in polar and devoid of charged amino acid residues. In mammals, compositionally similar domains modulate the assembly of dynamic stress granules (SGs) that associate via multivalent weak interactions. Dysregulation of SGs composed of proteins with prion-like domains has been proposed to underlie the formation of pathological inclusions in several neurodegenerative diseases. The events that drive prion-like domains into transient or solid assemblies are not well understood. We studied the interactors of the prototype prion domain NM of *Saccharomyces cerevisiae* Sup35 in its soluble or fibril-induced prion conformation in the mammalian cytosol. We show that the interactomes of soluble and prionized NM overlap with that of SGs. Prion induction by exogenous seeds does not cause SG assembly, demonstrating that colocalization of aberrant protein inclusions with SG components does not necessarily reveal SGs as initial sites of protein misfolding.

## Introduction

Prions of yeast and filamentous fungi constitute self-replicating entities composed of higher-order protein polymers. Depending on the genetic make-up and environmental factors, prions of lower eukaryotes can be harmful, benign, or even advantageous under selective pressure upon environmental changes (McGlinchey et al, 2011; Halfmann et al, 2012). Yeast prion induction is a rare event that can be triggered in response to environmental changes. In the prion conformation, fungal prion proteins exhibit stable cross-$\beta$ structures and self-perpetuating properties that allow them to persist over many cell generations by templating their own conformation onto soluble protein of the same kind. As such, yeast prions can be regarded as epigenetic determinants that store and transmit biological information to progeny and during mating. The translation termination factor Sup35, composed of the domains N, M, and C, represents the best-studied prion of *Saccharomyces cerevisiae* (King et al, 1997). Adoption of the prion conformation is a rare event and renders Sup35 inactive, resulting in translational readthrough and a change in metabolic phenotype. Conversion of Sup35 into the prion conformation can be templated by recombinant NM amyloid fibrils (King et al, 2006; Tanaka et al, 2006). The Sup35 N and M domains mediate the switch between the soluble functional and the insoluble prion state. The prion domain N is enriched in glutamines (Q) and asparagines (N) and is necessary and sufficient for establishment and maintenance of the prion conformation (Ter-Avanesyan et al, 1994; Derkatch et al, 1996). The charged middle domain M helps to keep the protein in its monomeric state (Glover et al, 1997), whereas the carboxyterminal C domain governs catalytic activity and is otherwise dispensable for prion formation (Glover et al, 1997).

Surprisingly, prions of lower eukaryotes share little to no sequence similarity with PrP$^{Sc}$ prions that cause transmissible spongiform encephalopathies in mammals. Instead, ~1% of the mammalian proteome contains proteins with low-complexity domains that compositionally resemble yeast prion domains (Alberti et al, 2009; Toombs et al, 2010; King et al, 2012). A characteristic feature of proteins with prion-like domains (PrlDs) is their ability to assemble into a variety of physiologically relevant membrane-less assemblies. Subcellular compartimentalization is driven by weak multivalent interactions that modulate liquid–liquid phase

[1]German Center for Neurodegenerative Diseases Bonn (DZNE e.V.), Bonn, Germany   [2]Bioinformatics and Genomics Programme, Centre for Genomic Regulation, Barcelona, Spain   [3]German Center for Neurodegenerative Diseases (DZNE), Munich, Germany   [4]Universitat Pompeu Fabra, Barcelona, Spain   [5]Institució Catalana de Recerca i Estudis Avançats, Barcelona, Spain   [6]Munich Cluster for Systems Neurology (SyNergy), Munich, Germany   [7]Neuroproteomics, School of Medicine, Klinikum rechts der Isar, and Institute for Advanced Study, Technical University of Munich, Munich, Germany   [8]Rheinische Friedrich-Wilhelms-Universität Bonn, Bonn, Germany

Correspondence: ina.vorberg@dzne.de
Verena Arndt's present address is Grünenthal GmbH, Aachen, Germany
Benedetta Bolognesi's present address is Institute for Bioengineering of Catalunya, Barcelona Institute of Science and Technology, Barcelona, Spain

separation. Examples are stress granules (SGs), highly dynamic organelles that rapidly and reversibly coalesce RNA-binding proteins and RNA under environmental stress (Kedersha & Anderson, 2007). Dysregulated assembly of membrane-less granules that results in aberrant protein aggregation and sequestration of vital cellular components has been implicated in the progression of neurodegenerative diseases (Wolozin, 2012). Recently, a group of RNA-binding proteins that take part in granule formation has been shown to form insoluble cytosolic or nuclear inclusions (Harrison & Shorter, 2017). Strikingly, several mutations in those genes associated with hereditary forms of neurodegenerative disorders are located in their PrlDs, strongly suggesting that these domains play a pivotal role in disease pathogenesis. The association of PrlD-containing proteins with aberrant protein inclusions and their role in ribonucleoprotein granule formation argues that the two processes might be somehow linked (Udan & Baloh, 2011). However, the molecular mechanisms that mediate liquid–liquid or liquid–solid transitions are not well understood.

The Q/N-rich PrlD of SG effector protein TIA-1 mediates liquid–liquid demixing, leading to the rapid formation of liquid-like droplets (Gilks et al, 2004). Interestingly, Sup35 NM can functionally replace the PrlD of TIA-1 and reconstitute its ability to form SGs (Gilks et al, 2004). Sup35 NM also exhibits bona fide prion activities in mammalian cells when expressed in the cytosol and exposed to recombinant NM amyloid fibrils (Krammer et al, 2009; Hofmann et al, 2013). Once induced, NM prions faithfully replicate over multiple cell divisions and induce self-sustained prion propagation in naive cells (Hofmann et al, 2013; Liu et al, 2016). Thus, a Q/N-rich PrlD can engage in the formation of two very different protein assemblies. To gain insight into interaction partners of PrlDs, we here used a systematic affinity tag purification and mass spectrometry approach and constructed host protein–protein interaction maps for Sup35 NM in its prion and non-prion state. We demonstrate that both soluble and prionized NM recruit RNA and RNA-binding proteins, including components of SGs. NM prions, however, are also under protein quality surveillance, as components of the protein degradation system associate with prionized NM. Importantly, exposure of cells to recombinant NM fibrils does not induce a SG response, demonstrating that SG proteins can become part of protein aggregates independent of SG formation. Thus, exogenous seeds can trigger the aggregation of homotypic PrlD-containing protein inclusions and result in the concomitant sequestration of SG components that are normal interactors of soluble PrlDs.

# Results

## Morphologically distinct NM prions propagating in individual N2a cell clones share comparable aggregate cores

We here aimed to analyze the interaction networks of an archetypical glutamine-/asparagine-rich prion domain in its soluble and aggregated, self-templating conformation in a mammalian cell environment. We used our established cell culture model that is based on the stable expression of the HA epitope-tagged Sup35 PrD (termed N) and its flanking domain (M) in the cytosol of mouse neuroblastoma cell line N2a (Krammer et al, 2009; Hofmann et al, 2013). The prion domain of the *S. cerevisiae* Sup35 shares no direct sequence homology to mammalian proteins, allowing us to explore consequences of its switch to the prion conformation without interfering with the normal cellular function of the protein. NM is soluble when expressed in the mammalian cytosol but can be induced to aggregate upon exposure of cells to recombinant NM fibrils (Krammer et al, 2009). Single cell clones had been isolated previously that propagate NM prions with strikingly different morphological and biochemical characteristics (Fig 1A) (Krammer et al, 2009). These cell clones faithfully replicate morphologically diverse NM aggregates over many cell divisions without loss of the aggregation phenotype (Krammer et al, 2009; Hofmann et al, 2013; Liu et al, 2016). N2a subclone 1c forms multiple small, fibrillar aggregates that often cluster in one region of the cell. N2a NM-HA cell clone 2e is characterized by a high number of small, punctate aggregates scattered throughout the cytoplasm, whereas subclone 3b forms large, worm-like aggregates. Importantly, persistent formation of NM-HA$^{agg}$ in N2a subclones is nontoxic (Hofmann et al, 2013). Filter trap analysis confirmed the presence of SDS-resistant NM-HA polymers in subclones 1c, 2e, and 3b, but not in cells expressing soluble NM-HA (N2a NM-HA$^{sol}$) (Fig 1B). NM-HA aggregates also persisted upon harsh denaturation and separation by semidenaturing detergent agarose gel electrophoresis (SDD–AGE), indicative of an amyloid-like structure of the NM-HA polymers present in clones 1c, 2e, and 3b (Fig 1C). Although NM-HA$^{sol}$ was clearly monomeric, most of the NM-HA was in its polymeric state in all three cell clones (Fig 1C). In line with NM-HA prions isolated from a bulk population of N2a cells (Duernberger et al, 2018), limited proteolysis with chymotrypsin demonstrated that the amino-terminus of NM-HA aggregates present in all cell clones was protease-sensitive, whereas a region spanning the octapeptide repeat region, the carboxyterminal N domain, and M were protected from proteolysis (Fig 1D–F). Thus, despite morphological differences, NM aggregates share a comparable region protected from proteolysis, probably comprising the aggregate core.

## Soluble and aggregated NM-HA sequester intrinsically disordered and nucleic acid–binding proteins

To identify NM aggregate interactors, NM-HA in its soluble and aggregated state was immunoprecipitated from cells and subjected to liquid chromatography–tandem mass spectrometry analysis. We performed three independent pull-downs for each cell line and analyzed each biological replicate by mass spectrometry twice. As a control, immunoprecipitation (IP) was also performed using anti-HA antibodies and wild-type N2a cells not expressing the transgene (Fig 2A). Proteins were classified as interactors if at least two unique peptides per protein were detected in at least two biological replicates. Interactors were excluded when two or more of the corresponding unique peptides were identified in one or more biological replicate of the negative N2a control. A total of 420 proteins was identified that bound to soluble and/or aggregated NM-HA but was not identified in the negative control (Fig 2B). Of the candidate interactors of total (soluble and aggregated) NM-HA, 5.5% were not found to interact with aggregated NM-HA in any of the

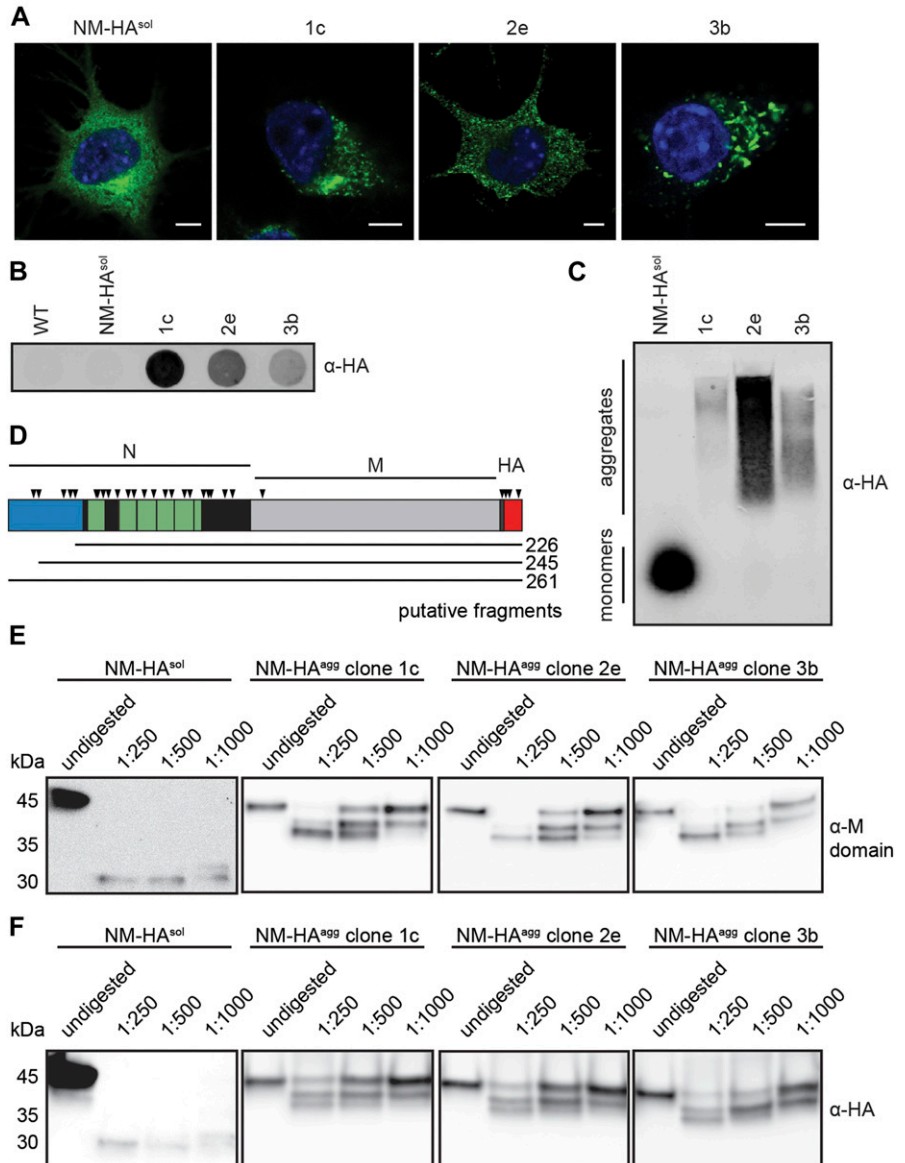

**Figure 1. Aggregation state of NM-HA in mouse neuroblastoma cell populations.**
**(A)** Immunofluorescence staining of mouse N2a neuroblastoma cells stably expressing soluble yeast Sup35 prion domain NM tagged with the HA antibody epitope (NM-HA$^{sol}$) and N2a subclones 1c, 2e, and 3b persistently producing NM-HA aggregates (NM-HA$^{agg}$). NM was detected using mAb anti-HA (green), and nuclei were stained with Hoechst (blue). Scale bar: 5 μm. Note that individual cell clones differ in their respective NM-HA expression levels (Krammer et al, 2009). Cell clones had been isolated from a N2a NM-HA bulk population exposed to recombinant NM amyloid fibrils. **(B)** The presence of SDS-resistant NM-HA in N2a subclones was determined by a filter trap assay. NM was detected using mAb anti-HA. **(C)** SDD–AGE analysis of lysates from N2a cells expressing NM-HA$^{sol}$ and NM-HA$^{agg}$. NM-HA was detected using mAb anti-HA. **(D)** Schematic illustration of NM-HA. A region highly abundant in Q/N at the amino-terminus is shown in blue (residues 1–38). Note that also other regions in N are enriched in Q/N. The octapepetide repeat region is depicted in green, and the carboxyterminal region of N is marked in black. Chymotrypsin preferentially cleaves at the carboxyl side of amide bonds of tyrosine, tryptophan, and phenylalanine (symbolized by arrow heads). Note that the M domain lacks these residues. Three tyrosine residues are present in the HA epitope. Putative fragments that correspond to the length of peptides identified by Western blot are indicated. **(E, F)** NM-HA aggregates derived from different N2a cell clones exhibit comparable chymotrypsin patterns. Lysates of N2a NM-HA$^{sol}$ and N2a NM-HA$^{agg}$ clones 1c, 2e, and 3b were subjected to increasing amounts of chymotrypsin or left untreated. Proteins were analyzed by SDS–PAGE and Western blot. NM was detected by 4A5 anti-M domain antibody raised against an epitope spanning amino acid residues 229–247 (Krammer et al, 2008b) (E) or mAb anti-HA (F) (both antibody-binding sites indicated in (D)).

clones (Fig. 2B). By contrast, approximately one-third (36%) of total identified proteins were interacting with both soluble and aggregated NM-HA (interactors of NM-HA$^{agg}$ of clones 1c, 2e, and 3b). Comparison of candidate NM-HA interactors revealed that 17% of proteins (73 proteins) were found associated with all aggregates types and soluble NM-HA (Fig 2C).

We assessed the physicochemical features of the proteins interacting with either NM-HA$^{agg}$ or NM-HA$^{sol}$ using the *clever-*Machine (CM) algorithm (Klus et al, 2014) to identify discriminative properties among NM-HA interactomes and random proteome data sets. Interactors of aggregated NM-HA derived from individual cell clones were pooled to create a large enough data set for analysis. Interactors of NM-HA$^{sol}$ and NM-HA$^{agg}$ were both enriched in intrinsic disorder and ability to bind nucleic acids (Fig 2D and E), features that are characteristic for proteins that take part in cytoplasmic granules, such as SGs (Fig 2F) (Jain et al, 2016). The

algorithm *cat*GRANULE (Bolognesi et al, 2016) further predicted that NM-HA$^{agg}$ and NM-HA$^{sol}$ interactors have a significantly higher ability to assemble into liquid-like granules than a random set of mouse proteins (Fig 2G). Gene ontology analysis confirmed that interactomes of NM-HA$^{sol}$ and NM-HA$^{agg}$ were enriched for proteins involved in RNA metabolism (Fig 3). Thus, morphologically diverse NM-HA aggregates propagating in different N2a cell clones recruit interacting partners involved in RNA metabolism. Comparison of the NM-HA interactomes with a curated list of SG proteins (Jain et al, 2016; Markmiller et al, 2018) revealed a significant overlap for soluble and aggregated NM-HA (Fig 4A and Table S1). 82% of the SG proteins interacting with NM-HA$^{sol}$ also associated with aggregated NM-HA (Fig 4B). Thirty-three SG proteins were found to interact with all NM-HA aggregates (Fig 4C). Consistent with the enrichment of the NM interactomes and the SG proteome for intrinsic disorder, proteins with PrLDs were substantially overrepresented in these interactomes compared with the entire human proteome (Fig 4D).

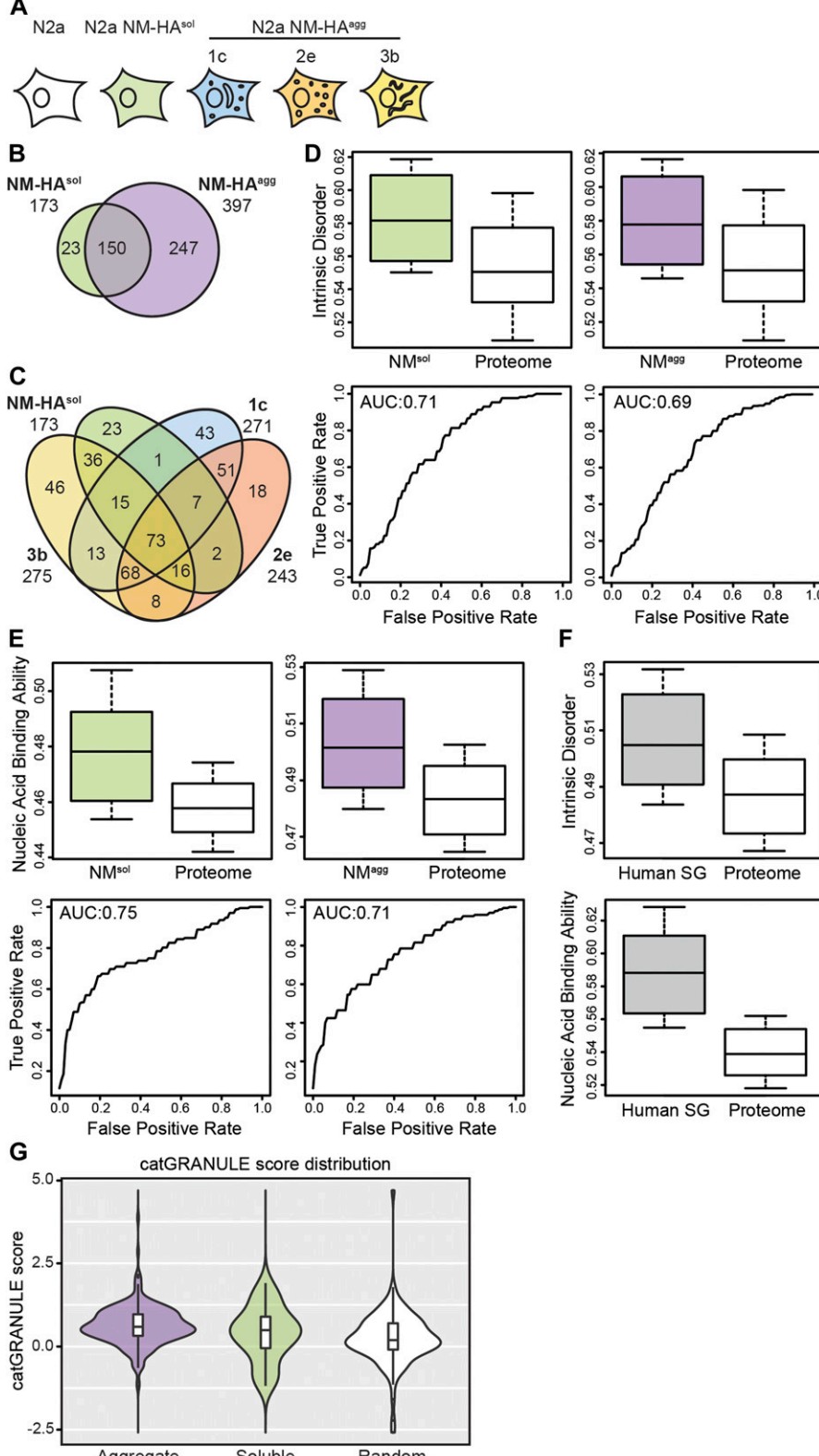

**Figure 2. Interactomes of soluble and aggregated NM-HA are enriched for intrinsically disordered and nucleic acid–binding proteins.**

**(A)** Analyzed cell lines. Biological triplicates of N2a subclones 1c, 2e, and 3b, N2a NM-HA[sol] and wild-type N2a cells were subjected to mass spectrometry analysis after SDS–PAGE and in-gel trypsin digestion of immunoprecipitated NM-HA using anti-HA antibodies. **(B)** Number of proteins found as putative interactors of soluble (NM-HA[sol]) and aggregated NM-HA (NM-HA[agg]) from cell clones 1c, 2e, and 3b combined. **(C)** Putative interactors of NM-HA[sol] and the NM-HA[agg] derived subclones 1c, 2e, and 3b individually. **(D)** Interactomes of soluble and aggregated NM-HA are enriched for intrinsically disordered proteins. Box-plot shows intrinsic protein disorder of interactors calculated using DisProt (Dunker et al, 2002). Intrinsic disorder of interactors was compared with that of a random subset of the mouse proteome. *P*-values are $1.7 \times 10^{-8}$ and $3.48 \times 10^{-9}$, respectively. Statistics were performed using the Kolmogorov–Smirnov test. Shown below is the corresponding area under the ROC curve (AUROC). **(E)** Interactomes of soluble and aggregated NM-HA are enriched for nucleic binding proteins. Box-plot displays the nucleic acid binding ability of interactors calculated using the scale of nonclassical RBD (Castello et al, 2012) compared with that of a random subset of the mouse proteome. *P*-values are $3.30 \times 10^{-13}$ and $3.46 \times 10^{-11}$, respectively (Kolmogorov–Smirnov test). Shown below is the AUROC. **(F)** Intrinsic disorder and increased nucleic binding ability are characteristics of SG proteins. Box-plot shows intrinsic protein disorder calculated (upper panel) using DisProt (Dunker et al, 2002) and nucleic acid binding ability (lower panel) calculated using the Castello scale (Castello et al, 2012) for SGs components compared with a random subset of the human proteome. **(G)** Interactors of NM-HA[agg] have a higher propensity to assemble into cytoplasmic granules compared with a random subset of the mouse proteome (Bolognesi et al, 2016). Interactors NM-HA[sol] versus NM-HA[agg]: *P* = 0.2973; NM-HA[sol] versus random proteome: *P* = 0.2101; NM-HA[agg] versus random proteome: $P = 2.348 \times 10^{-7}$. Statistics were performed using the Wilcoxon test.

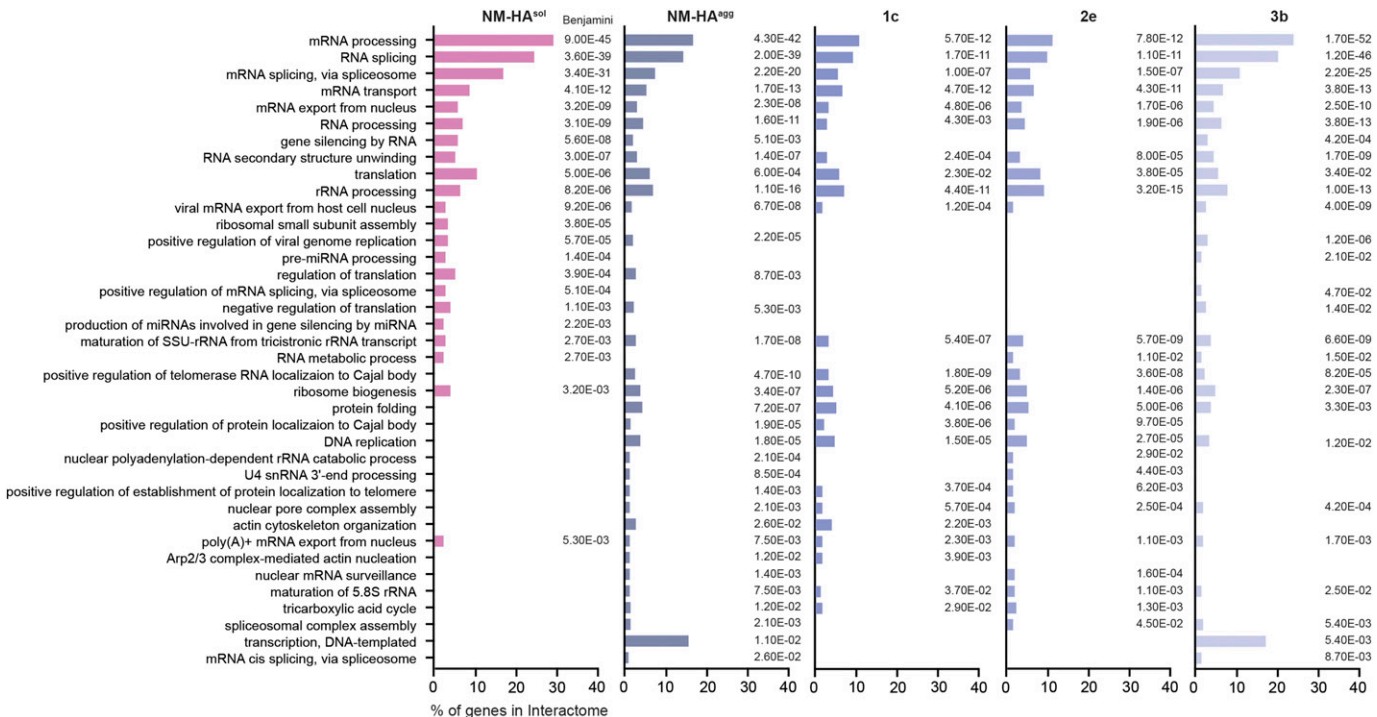

**Figure 3. Gene Ontology enrichment analysis of NM-HA interactomes.**
For all interactomes, the top 20 Gene Ontology biological process annotations were determined and combined in one list. Depicted are the percentages of genes that fall into this category compared with the respective interactome. Benjamini *P*-value is shown.

## NM prions share a subset of the SG interactome

The foregoing results suggested that NM-HA prions and SGs share a subset of interaction partners. To assess the general presence of SG markers in NM-HA aggregates, we used a bulk population of N2a cells stably producing SDS-resistant NM aggregates (Fig S1). Immunofluorescence staining of NM-HA[agg] cells revealed sequestration of small amounts of canonical SG marker TIA-1 by NM-HA aggregates (Fig 5A). IP of endogenous SG markers G3BP, TIAR, and TIA-1 also pulled down low amounts of NM-HA in cells replicating NM-HA prions (Figs 5B and S2A). We were unable to pull-down NM-HA in cells expressing soluble NM-HA, potentially because of lower sensitivity of Western blot analysis (Fig 5B). Consistent with the finding that RNA-binding proteins were present in interactor maps, traces of RNA were found in NM-HA prions (Fig 5C). Furthermore, immunofluorescence and IP revealed that SG proteins FUS and TDP-43 partially complexed with NM-HA prions (Fig S2B and D) and immunoprecipitated with aggregated NM-HA (Fig S2C and E). The association of NM with SG markers was confirmed using N2a cells expressing NM-GFP. In an N2a NM-GFP cell clone previously isolated that stably propagates NM-GFP prions induced by recombinant NM fibrils (Fig S3A) (Hofmann et al, 2013), NM-GFP[agg] associated with G3BP and transiently expressed Flag-FUS (Fig S3B–D). Interestingly, also soluble NM-GFP interacted with G3BP, consistent with our proteomic analysis of NM-HA[sol] interactors (Table S1). Thus, although we cannot exclude the possibility that some potential aggregate interactors bound to soluble NM-HA present in N2a NM-HA[agg] cells, confocal microscopy analysis clearly demonstrates that several SG markers and RNA associate with NM-HA aggregates.

Association of NM-HA[agg] with SG markers suggested that NM-HA could also be recruited to SGs. Indeed, after arsenite treatment of N2a cells expressing soluble NM-HA, NM-HA was found in newly formed SGs (Fig 5D). SGs are transient membrane-less assemblies that quickly dissolve once stress subsides (Kedersha et al, 2000; Kedersha & Anderson, 2007). We tested the effect of the association of NM with SGs by monitoring cells with SGs over several hours after arsenite challenge. As N2a cells did not tolerate well the exposure to 0.5 M arsenite for 1 h when subsequently cultured for prolonged periods of times, we switched to 30 min, a sublethal treatment for induction of SGs by arsenite (Kedersha et al, 2000). 1 h after arsenite treatment, N2a NM-HA[sol] cells contained SGs that co-stained for NM-HA (Fig S4A). However, SGs disappeared 2 h posttreatment, consistent with the transient nature of SGs (Kedersha et al, 2000). Similar results were obtained with N2a NM-GFP[sol] cells (Fig S4B–D). Quantitative analysis of N2a NM-GFP[sol] cells demonstrated the presence of NM-GFP–positive SGs 1 h after arsenite treatment and the complete absence of SGs 2 h postremoval of arsenite (Fig S4C and D). No TIAR/NM-GFP–positive puncta were observed 2–7 h posttreatment. We conclude that association of NM with SGs is transient and does not result in persistent SGs.

As fibril-induced NM-HA aggregates sequestered SG marker proteins and RNA, we wondered if translation was impaired in cells harboring NM prions. Consistent with stalled protein translation during SG formation (Kedersha & Anderson, 2007), aggregate-bearing cells also exhibited reduced protein synthesis compared with cells expressing soluble NM-HA (Fig 5E and F), without, however, affecting cell viability (Fig 5G). Thus, in analogy to SGs,

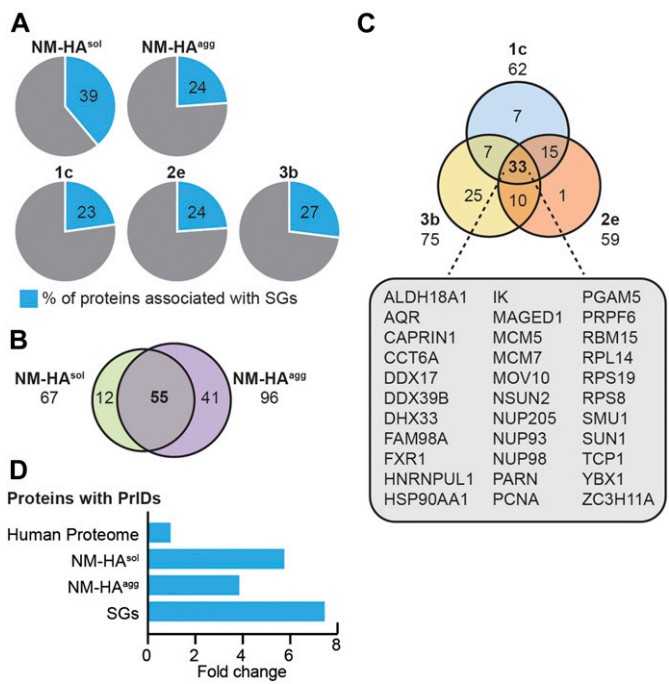

**Figure 4. Comparison of SGs and NM-HA interactomes.**
**(A)** Number of SG-associated proteins identified in the NM-HA interactomes. Interactomes were compared with a list of SG interactors curated from literature (Table S1) (Jain et al, 2016; Markmiller et al, 2018). Shown is the percentage of NM-HA interacting proteins that have been reported to be SG components. **(B)** Number of SG components interacting with both NM-HA[sol] and NM-HA[agg]. **(C)** Number of SG components associated with NM-HA[agg] in individual subclones 1c, 2e, and 3b (Table S1). **(D)** Enrichment of proteins with PrIDs in NM-HA[sol], NM-HA[agg], and SG interactomes (Table S1) compared with the human proteome (in fold change).

NM-HA prions recruit similar RNA-binding proteins, probably affecting overall protein synthesis.

### NM-HA prions are targeted by the cellular protein degradation machinery

Protein aggregates in metazoans are usually detected by cellular protein quality control and destined for degradation by autophagy (Bento et al, 2016). In line with this, components of the chaperone and autophagy system were found associated with aggregated NM-HA fractions by mass spectrometry. p62/Sqstm1, an adapter protein binding to ubiquitinated proteins destined to autophagic clearance, was identified as a putative interactor of all NM aggregate phenotypes (Table S1). IP of p62 successfully pulled down NM-HA in cell lysates of N2a NM-HA[agg] but not N2a NM-HA[sol] cells, demonstrating that p62 associated with the aggregated state of NM-HA (Fig 6A). Similar results were obtained in cells producing NM-GFP[agg] (Fig S3B and C). Colocalization of p62 with NM-HA[agg] or NM-GFP[agg] was further confirmed by immunofluorescence analysis (Figs 6B and S3E). Other putative NM-HA aggregate interactors were valosin containing protein (VCP), an AAA-ATPase that controls a large array of cellular functions, kelch-like ECH associated protein 1 (Keap1), which acts as a substrate adapter protein for an E3 ubiquitin ligase complex, and the molecular chaperone Ubiquilin-2. All three

proteins are involved in protein degradation via the ubiquitin–proteasome system and/or autophagy (Majcher et al, 2015). Confocal analysis of NM-HA[agg] cells revealed the sequestration of VCP to NM-HA aggregates, and the colocalization of Keap1 and Ubiquilin-2 with aggregated NM-HA (Fig 6B), arguing that NM prions are recognized by the protein quality machinery of the cell. No colocalization was observed with soluble NM-HA (Fig S5). Thus, NM prions sequester both components of SGs and the cellular protein quality control.

### NM prion induction by recombinant NM fibrils is independent of SG assembly

The sequestration of SG components by NM prions could be caused by recruitment of soluble NM and its interaction partners. Another possibility, however, was that exogenous NM fibrils triggered the assembly of SGs, which in turn served as initial sites of NM prion induction. We tested this possibility by monitoring SG assembly after exposure of N2a NM-HA[sol] cells to recombinant NM fibrils. Addition of NM fibrils to the medium induced NM-HA aggregation as soon as 1 h postexposure (Fig 7). However, NM fibrils did not trigger an SG response over a period of 24 h. The detection of canonical SG marker TIAR only at later stages after fibril addition (24 h) is probably because of a detection limit of less abundant proteins in smaller NM aggregates present at early stages after fibril addition. We conclude that recombinant fibril-induced NM prions form independently of SGs, suggesting that the sequestration of SG components by fibril-induced NM prions is driven by the passive recruitment of interactors of soluble NM into growing protein aggregates.

## Discussion

Here we studied the interaction network of the archetypical low-complexity Q/N-rich prion domain derived from the yeast Sup35 prion protein that lacks specific physiological function in mammalian cells. We demonstrate that the interactomes of both soluble and insoluble NM are highly enriched for intrinsically disordered proteins and RNA-binding proteins. Furthermore, we identified RNA as a component of NM-HA prions. Of note, RNA is present in infectious PrP-derived prions extracted from brains of animals infected with transmissible spongiform encephalopathies (Safar et al, 2005) and has been shown to trigger the formation of infectious PrP-derived prion particles in vitro (Wang et al, 2010). Many interactors of NM prion particles reside in networks of ribonucleoprotein complexes such as SGs. SGs contain repressed mRNA, ribosomal subunits, and other RNA-binding proteins (Jain et al, 2016; Markmiller et al, 2018), components also identified as putative interactors of Sup35 NM. The ability to associate with RNA-binding proteins was not dependent on the aggregation state or conformational differences of NM prions. This finding is in agreement with recent in vivo proximity-dependent biotinylation analysis demonstrating that interaction networks of canonical SG proteins remain largely unchanged under stress conditions, probably because SG components interact with similar proteins in submicroscopic granules or form transient interactions already under normal growth conditions (Youn et al, 2018).

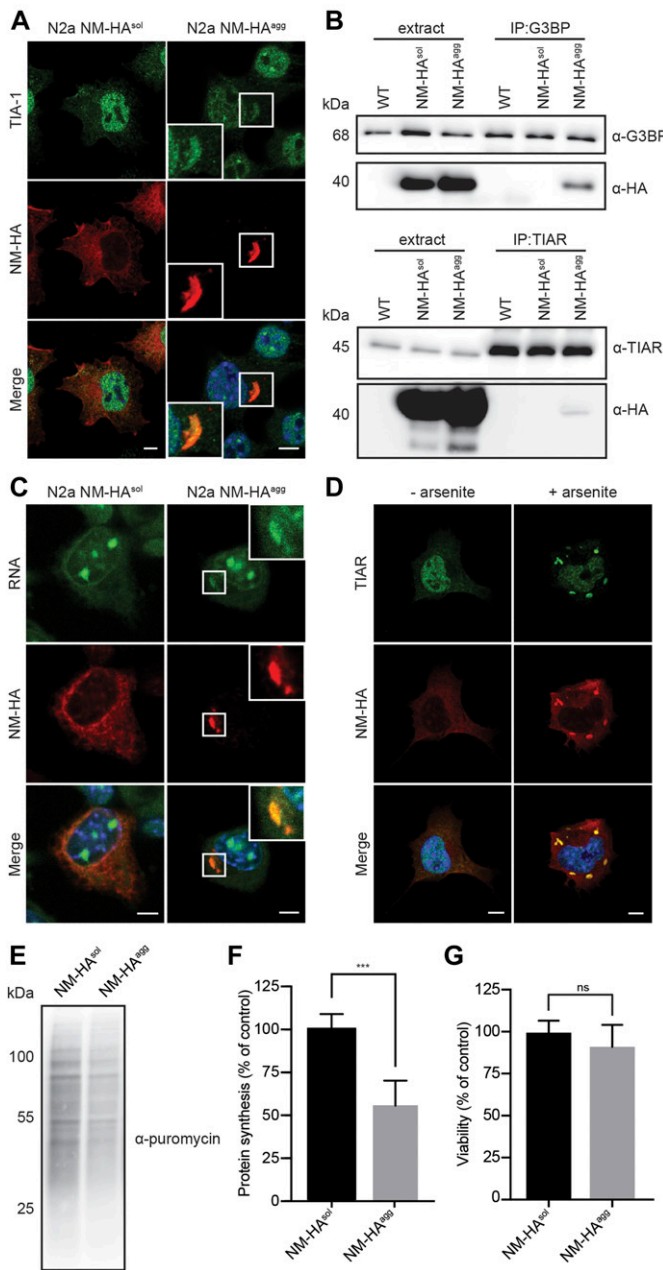

**Figure 5.  Shared components of NM-HA prions and SGs.**
**(A)** Immunofluorescence staining of N2a NM-HA[sol] and N2a NM-HA[agg] bulk cells. NM-HA was detected using mAb anti-HA (red) and SG marker TIA-1 was detected using pAb anti-TIA-1 (green). Nuclei were stained with Hoechst (blue). Scale bar: 5 μm. **(B)** IP of G3BP and TIAR from lysates of wild-type N2a, N2a NM-HA[sol], and N2a NM-HA[agg] bulk cells using mAb anti-G3BP or pAb anti-TIAR, followed by SDS–PAGE and Western blot. Total cell lysate (extract) was loaded as control. G3BP and TIAR were detected using mAb anti-G3BP and pAb anti-TIAR. NM-HA was detected using mAb anti-HA. **(C)** N2a NM-HA[sol] and N2a NM-HA[agg] cells were incubated with 1 μM SYTO RNASelect for 30 min and subsequently fixed with methanol, followed by immunofluorescence staining. NM was detected using mAb anti-HA (red), and RNA was visualized with SYTO RNASelect (green). Nuclei were stained with Hoechst (blue). Scale bar: 5 μm. **(D)** N2a NM-HA[sol] cells were treated with 0.5 mM sodium arsenite for 1 h to induce SGs or cells were left untreated. Immunofluorescence staining was performed using mAb anti-HA (red) and pAb anti-TIAR (green). Nuclei were stained with Hoechst (blue). Scale bar: 5 μm. **(E)** Protein synthesis was analyzed in N2a NM-HA[sol] and N2a NM-HA[agg] cells using the SUnSET method (Schmidt et al, 2009). Cells were incubated with

Under oxidative stress, soluble NM was also recruited to SGs. The finding that Sup35 NM interacts with SG components is in agreement with the sequestration of soluble Sup35 NM by arsenite-induced SGs in COS-7 cells (Gilks et al, 2004). Also in yeast, Sup35 interacts with RNA-binding proteins that are constituents of SGs (http://www.yeastgenome.org). Furthermore, ectopically expressed mammalian TIA-1 and its endogenous homolog Pub1 also associate with Sup35 NM in *S. cerevisiae* (Li et al, 2014). The partitioning of Sup35 NM into dynamic transient liquid condensates in vivo is driven by the prion domain (Franzmann et al, 2018; Khan et al, 2018). This is in agreement with in vitro binding of the Sup35 prion domain to hydrogels formed by SG components FUS or hnRNPA2, suggesting that NM directly interacts with heterologous PrlD-containing proteins (Kato et al, 2012).

Dysregulation of SG formation has been proposed to cause a gradual evolution of SGs into solid inclusions (Li et al, 2013; Patel et al, 2015). Support for the hypothesis that SGs convert into pathological inclusions comes from studies with recombinant PrlD-containing proteins that convert from hydrogels to fibrous structures over time in the test tube (Li et al, 2013; Patel et al, 2015). SG components are also constituents of pathologic inclusions formed by disease-associated proteins with PrlDs in neurodegenerative diseases such as amyotrophic lateral sclerosis or frontotemporal lobar degeneration (Liu-Yesucevitz et al, 2010). Furthermore, disease-associated mutations in PrlD-containing proteins affect the dynamics of the SG they are recruited to and delay, but do not inhibit, its dissolution (Mackenzie et al, 2017). We cannot exclude the possibility that the transition of PrlD-containing proteins to pathological aggregates is initiated in SGs in the absence of homologous amyloid seeds. However, our data demonstrate that SG components can be recruited to protein aggregates independent of SG induction. Once released into the cytosol, exogenous seeds likely sequester SG components, as these also interact with soluble NM that is recruited to the growing aggregate. The presence of SG components in fibril-induced NM prions that have been faithfully propagated over many cell divisions reflects the intrinsic ability of NM to associate with other SG proteins and RNA.

NM prions replicating in mammalian cells recruited a substantial number of proteins engaged in cellular protein quality control. Specifically, we found the ATP-dependent remodeling complexes CCT and RuvBL1/RuvBL2 associated with aggregated NM-HA. Studies in yeast recently demonstrated that the CCT complex inhibits SG assembly, whereas AAA+ ATPases RuvBL1 and RuvBL2 impair SG disassembly (Jain et al, 2016). Factors of both complexes have, however, also been found associated with aggresomes, perinuclear inclusions actively formed by the cell to sequester aberrantly folded proteins (Zaarur et al, 2015). In HeLa cells, a complex of RuvBL1 and RuvBL2 suppresses aggresome formation (Zaarur et al, 2015), suggesting that this mixed dodecamer

puromycin for 30 min. Incorporated puromycin was detected using mAb anti-puromycin (12D10). **(F)** Quantitative analysis of puromycin incorporation. Bars represent mean values ± SD. Statistical analysis was performed using *t* test (n = 3). Significant changes are indicated by asterisks (***$P ≤ 0.001$). **(G)** Viability of N2a NM-HA[sol] cells and N2a NM-HA[agg] cells was determined by XTT tetrazolium salt assay. Bars represent mean values ± SD. Statistical analysis was performed using *t* test (n = 3). Changes are not significant (ns).
Source data are available for this figure.

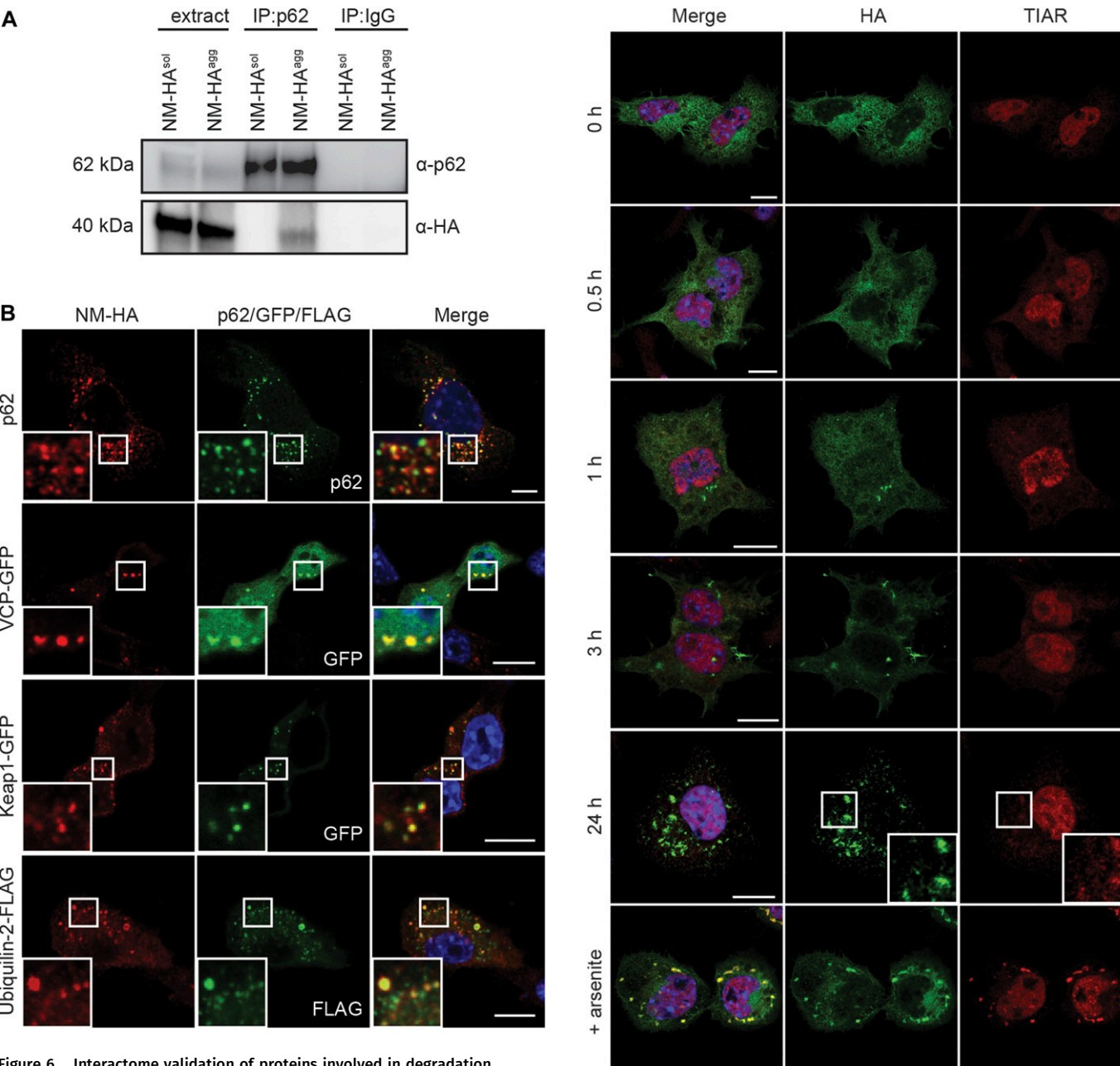

**Figure 6. Interactome validation of proteins involved in degradation pathways.**
**(A)** IP of endogenous p62 from cell lysates of wild-type N2a, N2a NM-HA[sol], and N2a NM-HA[agg] cells using mAb anti-p62, followed by SDS–PAGE and Western blot. Total cell lysate (extract) was loaded as control. p62 was detected using mAb anti-p62, and NM-HA was detected using mAb anti-HA. Pull-downs using unspecific IgG served as controls. **(B)** Bulk N2a NM-HA[agg] cells were either not transfected (for p62 detection) or transfected with constructs encoding for VCP-EGFP, Keap1-GFP, or Ubiquilin-2-FLAG and subjected to immunofluorescence staining 48 h posttransfection. NM-HA was stained using mAb anti-HA (red), p62 was detected using mAb anti-p62 (green), and FLAG was stained using mAb anti-FLAG (green). GFP is shown in green. Nuclei were stained with Hoechst (blue). Scale bar: 5 μm.

**Figure 7. Fibril-induced NM prions do not evolve from SGs.**
N2a NM-HA[sol] cells were incubated with 5 μM recombinant NM fibrils (monomer concentration) for the indicated time points and subsequently analyzed for NM-HA aggregate induction and SGs. As a positive control, cells were incubated with 0.5 mM sodium arsenite for 1 h without addition of fibrils. Immunofluorescence staining was performed using mAb anti-HA (green) and pAb anti-TIAR (red). Nuclei were stained with Hoechst (blue). Scale bar: 10 μm.

barrel-like complex plays an active role in both transient and stable protein assemblies. In *S. cerevisiae*, propagation and maintenance of most prion states is governed by disaggregase Hsp104 in conjunction with Hsp70 and Hsp40 that play a pivotal role in aggregate shearing (Chernoff et al, 1995). As Hsp104 has no homolog in the mammalian cytosol, it is unclear which cellular factors critically influence NM prion particle partitioning and dissemination in mammalian cells. Interestingly, Hsp70–Hsp40 powered by nucleotide exchange factor Hsp110 (also called Hsp105/HspH1) has recently been

shown to act as metazoan disaggregation machinery for misfolded protein aggregates but not for amyloid (Rampelt et al, 2012). Hsp110 is also involved in regulating NM fibril elongation in *S. cerevisiae* (O'Driscoll et al, 2015) and was found associated with both soluble and aggregated NM-HA in N2a cells. However, in vitro, Hsp110, Hsp70, and Hsp40 were unable to remodel Sup35 amyloid fibrils (Shorter, 2011), questioning if the same machinery can disassemble NM fibrils in vivo. In higher eukaryotes, protein aggregate disassembly is usually taken over by autophagy. Indeed, we found several autophagy markers associated with NM prions, including autophagy receptors p62, VCP, KEAP1, and Ubiquilin-2 (Majcher et al, 2015). Interestingly, autophagy has been implicated in the fragmentation of PrP-derived prions that cause fatal transmissible spongiform encephalopathies (Heiseke et al, 2010). However, VCP (Buchan et al, 2013) and autophagy receptor p62 (Matus et al, 2014) have also been reported to associate with SGs and to regulate SG dissolution. To what extent cellular protein quality control factors contribute to NM prion propagation remains to be investigated.

Aberrant association with host factors has been proposed to perturb cellular processes, thereby contributing to disease progression. Similar to the C-terminal aggregation-prone PrlD of TDP-43 (Chou et al, 2018), NM in its soluble and prion state recruits components of the nuclear core complex. It is unclear if the NM prion state also impairs cellular processes such as nuclear transport, as has been shown for aggregation of other PrlD-containing proteins, including TDP-43 (Prpar Mihevc et al, 2017; Chou et al, 2018). We previously demonstrated that NM prion propagation is not overtly toxic in mouse neuroblastoma cells or primary neuronal and glial cells (Hofmann et al, 2013). Still, NM prions reduce global protein synthesis, probably caused by the recruitment of RNA-binding factors. However, the fact that NM prions can be persistently propagated in mitotically active N2a cells over multiple cell passages argues that NM aggregation does not impose a negative selection on cell populations.

# Materials and Methods

## Chemicals

Chemicals were, if not specified otherwise, obtained from Sigma-Aldrich or Roth. Complete protease inhibitor was purchased from Roche. The Bradford protein assay was obtained from Bio-Rad, and the ECL chemiluminescence kit was from Pierce. mAb anti-HA and polyclonal antibody (pAb) anti-TIA-1 antibodies used for fluorescence microscopy were obtained from Santa Cruz Biotechnology; mAb anti-HA antibody used for IP was obtained from Roche; mAb anti-G3BP, pAb anti-FUS, pAb anti-TIA-1, pAb anti-TDP-43, and rabbit mAb anti-GFP ab183734 were purchased from Abcam; pAb anti-p62 was obtained from Progen; pAb anti-TIAR antibodies were purchased from Santa Cruz Biotechnology for IP and NEB for fluorescence microscopy; mAb anti-FLAG M2 antibody and mAb anti-puromycin clone 12D10 were purchased from Sigma-Aldrich. MAb 4A5 directed against the M domain of NM has previously been described (Krammer et al, 2008a). Fluorescein-conjugated secondary antibodies were purchased from Dianova or from Life Technologies.

## Cell lines

N2a cells expressing NM-HA and subclones with NM-HA aggregates (N2a NM-HA^agg) have been described previously (Krammer et al, 2009). N2a cells expressing soluble cytosolic NM-HA (N2a NM-HA^sol) were generated by lentiviral transduction. N2a NM-HA^agg clones 1c, 2e, and 3b cells were generated by exposing NM-HA^sol cells to 1 $\mu$M recombinant NM fibrils (monomer concentration) and subsequent selection of subclones stably producing NM-HA^agg of diverse phenotypes (Krammer et al, 2009). Cells were maintained in DMEM containing GlutaMAX (Gibco by Life Technologies) supplemented with 10% FCS (Biochrom) at 37°C in humidified air and 5% $CO_2$. N2a NM-GFP^sol and a subclone thereof producing NM-GFP^agg have been described (Hofmann et al, 2013). N2a cells stably expressing EGFP were generated by lentiviral transduction as previously described (Sachdev et al, 2018). Cells were kept in Opti-MEM (Gibco by Life Technologies) and 10% FCS.

## Transfection

N2a cells were transfected with Effectene or Lipofectamine 2000 Transfection Reagent (QIAGEN) according to the manufacturer's instructions. All plasmids used were obtained from Addgene. pcDNA5/FRT/TO 3× FLAG-FUS WT was generated by the Reed laboratory (Yamazaki et al, 2012), pDONR TDP43 WT YFP was a gift from Aaron Gitler's laboratory (Johnson et al, 2008), and p4455 FLAG-hPLIC-2 was generated by Peter Howley (Kleijnen et al, 2000). For expression in mammalian cells, the open reading frame was cloned into pcDNA3.1/zeo (+). pEGFP-VCP was generated by Nico Dantuma (Tresse et al, 2010) and phrGFP-Keap1 by Qing Zhong (Fan et al, 2010).

## Filter trap assay

Cell pellets were lysed in RIPA buffer (25 mM Tris–HCl, pH 8, 150 mM NaCl, 0.1% SDS, 0.5% sodium deoxycholate, 1% Nonident P-40 [NP-40], 10% glycerol, 2 mM EDTA, and complete protease inhibitor) for 30 min on ice. Cell lysates were sonicated (100% power, setting 1), and samples were adjusted to comparable protein concentrations. 20–50 $\mu$g of total protein was vacuum transferred onto nitrocellulose membrane (pore size 0.2 $\mu$m). Wells were rinsed four times with SDS wash buffer. The membrane was washed in TBST and blocked with 5% nonfat dry milk in TBST. After incubation with primary and IRDye 800CW (LI-COR)-conjugated secondary antibodies, the membrane was scanned using an Odyssey imager (LI-COR).

## SDD–AGE

SDD–AGE was performed according to a previously published protocol (Halfmann & Lindquist, 2008). Briefly, cells in a six-well plate were rinsed in PBS and lysed (50 mM Tris–HCl, pH 7.5, 150 mM NaCl, 1% NP-40, and complete protease inhibitor). Cellular debris was pelleted by centrifugation (500*g*, 2 min), and supernatant was mixed with sample buffer (0.5× Tris-acetate-EDTA [TAE], 5% glycerol, 2% SDS, and bromphenol blue). Samples were incubated at RT for 5 min and loaded onto 1.5% agarose, 0.1% SDS gels (TAE). The gel was

run at 9–10 V overnight at 4°C. Proteins were transferred onto nitrocellulose membrane by capillary transfer and detected using primary and secondary antibodies as described.

### IP of NM-HA and NM-GFP complexes

Confluent monolayers in 10-cm dishes were rinsed in PBS and lysed (50 mM Tris–HCl, pH 7.5, 150 mM NaCl, 0.5% NP-40, and complete protease inhibitor). Lysates were transferred to low-bind tubes (Eppendorf) and incubated for 30 min at 4°C. Cell debris was pelleted by low-speed centrifugation (500$g$, 2 min). Cytosolic protein extracts were adjusted to comparable protein concentrations and incubated with indicated antibodies overnight at 4°C, on a rotating wheel. Dynabeads (Life Technologies) were added and samples were rotated for 1 h at 4°C. Tubes were placed in a magnetic rack (Life Technologies), and captured protein complexes were rinsed five times in lysis buffer. Beads were resuspended in sample buffer, and proteins were eluted by boiling for 5 min at 95°C. Bead-free samples were transferred to new reaction tubes. For isolation of proteins bound to NM-GFP, GFP-trap magnetic beads (ChromoTek) were used according to the manufacturer's instructions.

### Puromycin treatment

Cells were incubated with 10 $\mu$g/ml puromycin for 30 min, subsequently harvested and lysed (50 mM Tris–HCl, pH 7.5, 150 mM NaCl, 0.5% NP-40, and complete protease inhibitor). Samples were analyzed by SDS–PAGE and Western blot.

### Western blot analysis

Samples were mixed with 2× sample buffer (Bio-Rad) and denatured at 95°C for 5 min. Proteins were separated on 4–12% gradient precast gels (Bio-Rad). Proteins were transferred to nitrocellulose membranes. Membranes were blocked in 5% nonfat dry milk in TBST und subsequently incubated with primary antibodies and respective horseradish peroxidase–conjugated secondary antibodies (Dianova). Proteins were detected using the enhanced chemiluminescence reagent ECL (Pierce), and signals were captured using the STELLA 3200 imaging system (Raytest).

### Chymotrypsin proteolysis

Cells were lysed for 30 min at 4°C (50 mM Tris–HCl, pH 7.5, 150 mM NaCl, 0.5% NP40), and cell debris was pelleted by centrifugation (500$g$, 2 min). Lysates were mixed with chymotrypsin at protein to protease ratios between 1:250 and 1:1,000 and incubated on ice for 1 h. The reaction was terminated by addition of Pefabloc and 4× SEB buffer (Life Technologies). Proteins were analyzed by Western blot using 12% NuPage Bis-Tris gels (Life Technologies).

### Cell viability by XTT assay

Cells were seeded on 96-well plates, and after 24 h, XTT assay was performed. XTT labeling mixture was added to cells and incubated for 4 h in a humidified atmosphere (37°C, 5% $CO_2$). Absorbance was measured using a FLUOstar Omega microplate reader (BMG Labtech).

### Fluorescence microscopy

Cells grown on cover slips were rinsed with PBS and fixed with 4% paraformaldehyde at RT for 10 min. For antigen TDP-43 retrieval, cells were incubated with 6 M guanidine hydrochloride for 15 min after fixation. Cells were subsequently incubated with 0.5% Triton X-100 for 10 min and blocked with 2% goat serum (Dianova) for 1 h. After three washes with PBS, proteins of interest were bound using primary antibodies. After rinsing, cells were incubated with the suitable fluorophore-conjugated secondary antibodies at RT for 1 h. Cellular nuclei were stained with Hoechst (Molecular Probes) using a 1 $\mu$g/ml dilution in PBS for 5 min at RT. Slides were mounted in Aqua Poly/Mount (Polysciences). Images were captured using a LSM 700 or LSM 800 confocal laser scanning microscope (Zeiss). For staining of RNA, life cells were incubated with 1 $\mu$M Syto RNASelect (Life Technologies) for 20 min at 37°C. Cells were rinsed two times with PBS and subsequently fixed with ice cold methanol for 10 min at –20°C. For quantification of cells carrying aggregates, cells were seeded on 96-well plates and after 24 h subjected to immunofluorescence staining using anti-HA antibody. Images were captured using the Cell Voyager 6000 (Yokogawa) and analyzed using Cell Profiler.

### Bioinformatic analysis

The analysis of the different protein data was performed using the CM algorithm. The CM algorithm analyzes physicochemical properties of two protein datasets. The tool creates profiles, or physicochemical signatures, for each protein, taking into account a large set of features—both experimentally and statistically derived from other tools. Ten propensity predictors are used for each property, and the best performing one is reported in a box-plot. Further information can be found at http://s.tartaglialab.com/page/clever_suite (Klus et al, 2014). Protein–protein interactions reported in BioGRID (Stark et al, 2006), Reactome (Croft et al, 2011), and NCI-PID (Schaefer et al, 2009) were used to quantify the protein networks affected by co-sequestration and changes in protein–protein interactions of soluble and insoluble NM-HA.

### Gene Ontology analysis

Gene Ontology analysis was performed using the DAVID bioinformatics tool (https://david.ncifcrf.gov/home.jsp). The top 20 biological process annotations for each interactome were chosen.

### Liquid chromatography–tandem mass spectrometry

A shotgun strategy was applied to identify interaction partners of NM-HA aggregates. Three biological replicates of wild-type (wt) N2a, N2a NM-HA[sol], and N2a cell clones 1c, 2e, and 3b harboring NM-HA[agg] were collected for quantitative proteomics analysis. To this end, cells from three near-confluent 15-cm dishes per cell population were scraped into lysis buffer (50 mM Tris–HCl, pH 7.5, 150 mM NaCl, 0.5% NP-40, 1× PhoSSTOP phosphatase inhibitor, and 1× complete protease inhibitor). The protein concentration was determined via

Bradford and adjusted to 10 mg/ml. NM-HA was immunoprecipitated as above. Immunoprecipitated proteins were eluted in Laemmli buffer and separated on 4–12% gradient precast gels (Bio-Rad). Gels were stained with colloidal Coomassie stain (Bio-Rad). Each protein lane was cut into 10–11 pieces and subjected to in-gel trypsin digestion as previously described (Shevchenko et al, 2007). Fractions of individual samples were pooled, and peptides were separated on a nanoLC system (EASY-nLC II, Proxeon—part of Thermo Fisher Scientific) using an in-house packed C18 column (fused silica 15 cm × 75 $\mu$m ID, New Objective; ReproSil-Pur 120 C18-AQ, 2.4 $\mu$m, Dr. Maisch GmbH) with a binary gradient of water and acetonitrile (ACN) containing 0.1% formic acid (Exp 1: 0 min, 8% ACN; 145 min, 23% ACN; 225 min, 42% ACN; 227 min, and 95% ACN; Exp 2 + 3: 0 min, 10% ACN; 80 min, 25% ACN; 115 min, 42% ACN; and 116 min, 95% ACN). The nanoLC was coupled online to a Velos Pro Orbitrap mass spectrometer via a Nanoflex ion source (Thermo Fisher Scientific). Full-scan MS spectra were acquired using the orbitrap mass analyzer with a resolution of 60,000, a mass range from 300 to 2,000 m/z and a target value of 1,000,000 counts. The 14 most intense peptide ions were chosen for collision-induced dissociation within the ion trap (target 10,000 counts, threshold 1,000 counts, isolation width 2 m/z, normalized collision energy 35%, activation Q 0.25, and activation time 10 ms).

### Data analysis

The MS data were analyzed using the Maxquant search algorithm (maxquant.org, Max-Planck Institute Munich, version 1.5.3.12) (Cox et al, 2014). The MS raw data were searched against a canonical FASTA database of *Mus musculus* from UniProt including the sequence of the HA-tagged NM protein (download: 2016-04-11; 50,721 entries). Trypsin was defined as protease. Two missed cleavages were allowed. The option first search was used to recalibrate the peptide masses within a window of 20 ppm. For the main search peptide and peptide fragment, mass tolerances were set to 4.5 ppm and 0.5 D, respectively. Carbamidomethylation of cysteine was defined as static modification. Acetylation of the protein N-terminal and oxidation of methionine was set as variable modifications. The false discovery rate for both, peptides and proteins, was adjusted to less than 1% using a target and decoy database approach. Label-free quantification of proteins required at least two ratio counts of razor or unique peptides.

## Supplementary Information

## Acknowledgements

We thank Dr. Reed and Dr. Gitler for depositing constructs at Addgene. We thank Meike Brömer and Della David for critical input on this article. We thank Ireen König for advice on confocal microscopy. This work was supported by Centers of Excellence in Neurodegeneration and by the Deutsche Forschungsgemeinschaft (German Research Foundation) within the framework of the Munich Cluster for Systems Neurology (EXC 2145 SyNergy).

## Author Contributions

K Riemschoss: conceptualization, data curation, formal analysis, validation, investigation, visualization, methodology, and writing—original draft.

V Arndt: conceptualization, data curation, formal analysis, validation, investigation, visualization, methodology, and writing—review and editing.

B Bolognesi: software, formal analysis, and writing—review and editing.

P von Eisenhart-Rothe: data curation, investigation, and visualization.

S Liu: data curation, investigation, and visualization.

O Buravlova: data curation and investigation.

Y Duernberger: investigation, methodology, and writing—review and editing.

L Paulsen: investigation and methodology.

A Hornberger: data curation, investigation, and visualization.

A Hossinger: data curation and investigation.

N Lorenzo-Gotor: software, formal analysis, and writing—review and editing.

S Hogl: software, formal analysis, and writing—review and editing.

SA Müller: software, formal analysis, and writing—review and editing.

G Tartaglia: software, formal analysis, supervision, and writing—review and editing.

SF Lichtenthaler: software, formal analysis, supervision, funding acquisition, and writing—review and editing.

IM Vorberg: conceptualization, supervision, writing—original draft, and project administration.

## Conflict of Interest Statement

The authors declare that they have no conflict of interest.

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
