## [Reviewer comments · Life Science Alliance]

Life Science Alliance

Fibril-induced glutamine-/asparagine-rich prions recruit stress granule proteins in mammalian cells

Katrin Riemschoss, Verena Arndt, Benedetta Bolognesi, Philipp von Eisenhart-Rothe, Shu Liu, Oleksandra Buravlova, Yvonne Duernberger, Lydia Paulsen, Annika Hornberger, Andre Hossinger, Maria de las Nieves Lorenzo Gotor, S. Hogg, Stephan Mueller, Gian Tartaglia, Stefan Lichtenthaler, and Ina Vorberg

DOI: <https://doi.org/10.26508/lsa.201800280>

Corresponding author(s): Ina Vorberg, German Center for Neurodegenerative Diseases (DZNE)

Review Timeline:

Submission Date:	2018-12-14
Editorial Decision:	2019-01-24
Revision Received:	2019-06-05
Editorial Decision:	2019-06-21
Revision Received:	2019-06-22
Accepted:	2019-06-25

Scientific Editor: Andrea Leibfried

Transaction Report:

January 24, 2019

Re: Life Science Alliance manuscript #LSA-2018-00280

Dr. Ina Vorberg
German Center for Neurodegenerative Diseases (DZNE)
Cell Biology and Pathophysiology of Prions
Sigmund-Freud-Strasse 27
Bonn 53127
Germany

Dear Dr. Vorberg,

Thank you for submitting your manuscript entitled "Fibril-induced glutamine-/ asparagine-rich prions sequester components of stress granules in mammalian cells" to Life Science Alliance. The manuscript was assessed by expert reviewers, whose comments are appended to this letter.

As you will see, the reviewers appreciate your work and think that it provides avenues for future research. However, they also note some technical shortcomings that would need to get addressed for publication here. We would thus like to invite you to submit a revised version, addressing the specific concerns raised by the reviewers. Importantly, proper controls (control IP with extracts from a cell line containing NM aggregates devoid of the HA tag; ref#3, point 1) and additional tests (reversibility of aggregation process (Ref#1, point 2); Venn diagram (Ref#1, point 1)) as well as further testing of the hypothesis put forward (ref#3, point 3) should be performed. While using a similar approach with a mammalian aggregate would add further value (see comments of ref#2 and #3), all reviewers concluded during the cross-commenting session that inclusion of such parallel approach is not needed at this stage.

Thank you for this interesting contribution to Life Science Alliance. We are looking forward to receiving your revised manuscript.

Sincerely,

- A letter addressing the reviewers' comments point by point.
- An editable version of the final text (.DOC or .DOCX) is needed for copyediting (no PDFs).
- High-resolution figure, supplementary figure and video files uploaded as individual files: See our detailed guidelines for preparing your production-ready images, <http://life-science-alliance.org/authorguide>
- Summary blurb (enter in submission system): A short text summarizing in a single sentence the study (max. 200 characters including spaces). This text is used in conjunction with the titles of papers, hence should be informative and complementary to the title and running title. It should describe the context and significance of the findings for a general readership; it should be written in the present tense and refer to the work in the third person. Author names should not be mentioned.

B. MANUSCRIPT ORGANIZATION AND FORMATTING:

Full guidelines are available on our Instructions for Authors page, <http://life-science-alliance.org/authorguide>

Reviewer #1 (Comments to the Authors (Required)):

Many proteins in the human protein contain so-called Prion-like Domains (PrID): low complexity domains responsible for the formation of membrane-less organelles, such as stress granules (SGs). It has been proposed that dysregulation of such SGs can result in pathogenic protein aggregation, however, the molecular mechanisms mediating this process remains elusive. Here, the authors use the NM prion domain of the yeast Sup35 prion protein, which shows a remarkably similar amino acid composition to PrID-containing proteins, as a model protein to study this mechanism in a mammalian background. They found that the NM prion (NM-HAagg) as well as the soluble NM (NM-HAsol) interactome are enriched for intrinsically disordered proteins, RNA-binding proteins and RNA, a highly similar composition as observed for SGs. Importantly, in contrast to the previously proposed mechanism of PrID-containing proteins, where aberrant SG formation upon certain stress conditions can lead to formation of pathogenic inclusion, the authors here conclude that these proteins can be recruited to inclusions independent of SG formation.

1. Fig 4A shows the percentage of identified SG proteins in both groups (NM-HAsol and NM-HAagg) is roughly the same, however the identity of these proteins is not mentioned here. For clarity and to support the suggested hypothesis, the authors should include a Venn diagram of all SG proteins identified in both NM-HAsol and NM-HAagg interactomes. If the hypothesis is correct, and the recruitment of SG proteins in aggregates is a result of the interaction of these proteins with soluble NM, most proteins should be present in both groups.

2. Upon oxidative stress, soluble NM was recruited into SGs (Fig 5D). The authors should show that this is a reversible process to exclude the possibility that NM aggregation occurs as a result of this stress condition.

3. The SG proteins TIA-1 and TIAR, which clearly bind NM-HAagg in the in vitro pull-down and immunostaining assays, are not detected by the LC-MS/MS approach. The authors should elaborate on why they used these two proteins to perform these control experiments, instead of other SG components that are identified by the LC-MS/MS approach.

Reviewer #2 (Comments to the Authors (Required)):

The interesting article by Riemschoss and co-workers studies the effect of heterologous expression in cultured mouse cells of a Q/N-rich prion domain (NM) from yeast Sup35. Using cell biology (immuno-fluorescence) and systems (proteomics: affinity tag purification followed by mass spectrometry) approaches, the authors compare the portion of the mouse proteome that co-purifies with the yeast sequences, both in its soluble or aggregated states. The proteins identified were then compared with those characteristics of membrane-less stress (P)-granules. Transfection of in vitro-assembled NM fibrils, which templates the assembly of prion aggregates in the recipient cells forming two distinct lineages (in terms morphology and toxicity), does not result in induction of stress granules.

The article, although merely descriptive in a good part, is interesting. It builds on previous results by the same group reporting the successful propagation of the same yeast prion (Sup35) in mammalian cells, by contributing now the clue that amyloid aggregation does not necessarily imply the assembly of stress granules: the same proteins (many of them RNA binding proteins, such as TDP-43 and FUS, also carrying prion-like domains) that usually form these membrane-less organelles were identified in their analysis as sequestered at the prion-like aggregates. These

aggregates also include proteins implicated in processes of protein quality control (proteostasis), tagging aggregates for autophagy and degradation at the proteasome.

The experimental evidence provided is technically sound and the manuscript deserves publication. However, the only doubt is whether the conclusions extracted from such heterologous studies can be fully translated into the natural, homotypic mammalian system. As the authors state at the Discussion section, the metazoan Hsp110/70/40 machinery that disassembles protein aggregates in the cytosol is unable to do so on those formed by Sup35 (NM). In addition, the same authors had shown that this yeast prion is not cytotoxic when expressed in mammalian cells. The answer should come from complementary experiments not transfecting yeast prion aggregates, but aggregates from a disease-relevant mammalian protein expressed in the recipient cells.

Reviewer #3 (Comments to the Authors (Required)):

In the manuscript entitled "Fibril-induced glutamine-/ asparagine-rich prions sequester components of stress granules in mammalian cells" Riemschoss and colleagues studied the interaction partners of an HA-tagged NM domain (NM-HA) of the yeast prion Sup35 in mouse neuroblastoma (N2a) cells. They employed extracts prepared from stable cell lines containing either soluble or aggregated NM for an immunoprecipitation with anti-HA antibodies. The proteins present in the immunopellet were then identified by LC-MS/MS. This analysis revealed that the interactomes of soluble and 'prionized' NM-HA overlap with that of stress granules.

In my opinion the major shortcoming of the manuscript is that it is very descriptive with very little mechanistic insights. For example, one would like to see that the putative interaction of NM-HA with any of the identified proteins has functional consequences. In addition, it is difficult to appreciate that the analysis of the interactome of a yeast protein in mammalian cells will help to enhance our knowledge about physiological or pathophysiological activities of mammalian prion or prion-like proteins.

- Throughout the manuscript the authors discuss their findings in the context of the mammalian prion protein (PrP) or mammalian RNA-binding proteins with low complexity prion-like domains. What is the advantage of an artificial system - the interactome of a yeast protein in mammalian cells - instead of analyzing directly interactors of either PrP or prion-like domains of the respective RNA-binding proteins?

- The authors compare the immunopellets of a protein in two different states, soluble and aggregated. There are some technical problems with such an approach:

1. They have a control for unspecific interactions of the antibodies/beads in extracts containing the soluble NM-HA, however an appropriate control is missing for the extracts containing aggregated/'prionized' NM-HA. Thus, it remains to be proven that proteins identified in the N2a-NM-HAagg cells extracts are indeed specific interactors of the aggregated NM-HA.
2. The IP is critically dependent on the accessibility of the HA tag. As a consequence only a small subset of aggregated NM-HA with an exposed HA tag, maybe with very distinct and not representative properties, is immunoprecipitated.
3. N2a-NM-HAagg cells contain in addition to aggregated NM-HA also soluble NM-HA. This is the fraction after translation and before recruitment to NM-HA aggregates. Therefore, one has to consider that at least some of 'common' interactors identified N2a-NM-HAagg cells may have not interacted with aggregated NM-HA.

Dr. Andrea Leibfried
Executive Editor
Life Science Alliance

Bonn, 05.06.19

Dear Dr. Leibfried,

Thank you for sending our manuscript entitled "Fibril-induced glutamine-/ asparagine-rich prions sequester components of stress granules in mammalian cells", now revised title "Fibril-induced glutamine-/asparagine-rich prions recruit stress granule proteins in mammalian cells" for review. We would like to thank the reviewers for their thoughtful and constructive comments and appreciate the invitation to submit a revised version. We have now performed extensive additional experiments to address the reviewer's suggestions. We have included control experiments demonstrating that also soluble and aggregated NM tagged with GFP interact with components of stress granules (SGs) and the quality control machinery. Further, we have followed the fate of stress granules and NM sequestered by SGs over a time period of 7 h post arsenite treatment and show that this sequestration does not lead to persistent SGs. As requested, we have also included a Venn diagram that demonstrates that stress granule interactors of soluble and aggregated NM strongly overlap.

We hope that the changes we made to the manuscript will make it suitable for publication in *Life Science Alliance*.

Sincerely,
Ina Vorberg, PhD

Point-by-point response to the comments:

Response to Editor's comments

Importantly, proper controls (control IP with extracts from a cell line containing NM aggregates devoid of the HA tag; ref#3, point 1)

Response: We have now included supplementary Fig. S3 demonstrating that NM tagged with GFP also interacts with SG marker G3BP and autophagy protein p62. Please note that we were unable to solely use untagged NM as this protein exhibits poor antigenicity in its aggregated form due to amyloid core formation of nearly the complete NM sequence in aggregated NM (see Figure 1E, F and (Duernberger et al., 2018)) and low affinity of our NM antibody.

and additional tests (reversibility of aggregation process (Ref#1, point 2);

Response: We have now assessed the reversibility of NM-positive SGs over time in both N2a cells expressing NM-HA as well as in N2a cells expressing NM-GFP (see new supplementary Fig. S4). We demonstrate that SG formation in cells expressing NM-HA^{sol} or NM-GFP^{sol} is transient.

Venn diagram (Ref#1, point 1))

Response: We have included a Venn diagram to demonstrate the strong overlap of SG proteins associating with both NM-HA^{sol} and NM-HA^{agg} (revised Figure 4B).

Further testing of the hypothesis put forward (ref#3, point 3) should be performed.

Response: Please see our detailed response below. We agree with the reviewer that we cannot exclude the possibility that some of our identified interactors only interact with soluble NM protein. However, this is a problem that any study dealing with aggregated proteins is facing. We find that soluble NM associates with RNA-binding proteins and proteins associating with SGs. Recent studies have clearly demonstrated that RNA-binding proteins and SG proteins exist in submicroscopic complexes already under non-stress conditions and these assemblies can be immunoprecipitated (Youn et al., 2018).

To reduce chances that interactors bound solely to non-aggregated NM, we used three cell clones that had been selected because they stably propagate fibril-induced aggregates in progeny cells (Hofmann et al., 2013; Krammer et al., 2009). We demonstrate the polymeric state of NM-HA^{agg} in these clones by SDD-AGE. Major findings of our mass spec analyses were that NM-HA in its aggregated state recruits components of SG and the protein quality control system. We validated these findings by confocal microscopy, demonstrating co-localization of aggregated NM-HA with several SG components, RNA and components of the quality control system. We have now also included additional experiments in which we demonstrate interaction and co-localization of aggregated GFP-tagged NM with putative interactors in a cell clone selected for production of NM-GFP aggregates in virtually all cells (new supplementary Fig. S3). Thus, our conclusion that NM interacts with components of SGs and the quality control machinery is valid. We now acknowledge the possibility that some interactors might only bind to soluble NM on page 11.

While using a similar approach with a mammalian aggregate would add further value (see comments of ref#2 and #3), all reviewers concluded during the cross-commenting session that inclusion of such parallel approach is not needed at this stage.

Response: We agree with the reviewers that repetition of our study with mammalian proteins with prion-like domains will be interesting. Aim of our current study was to identify interactors of an archetypal prion-like domain in the mammalian context that is non-toxic and behaves as a bona fide prion. We chose to use Sup35 NM that can form stable prions in mammalian cells (Krammer et al., 2009), allowing us to select cell clones as a source of faithfully propagating prions (or “prion-like” aggregates, as they

constitute artificial prions in mammals). To the best of our knowledge, this is the only prion-like protein for which **continuous** propagation of induced aggregates has been demonstrated in mammalian cells so far. While, for example, cellular models for TDP-43 aggregate induction exist, it is unclear if TDP-43 in these cellular models exhibits all prion characteristics (Nonaka et al., 2013; Smethurst et al., 2016). A hallmark of prions is their propagation by secondary nucleation such as fragmentation to produce more infectious seeds. This can be monitored experimentally for mammalian PrP-derived prions and NM prions by continuous propagation in mitotically active cells (Ghaemmaghami et al., 2007; Krammer et al., 2009; Krauss and Vorberg, 2013). Protein aggregates that cannot propagate, for example those composed of a Huntingtin fragment, are not fragmented and thus not inherited (Rujano et al., 2006). Further, NM prions are naturally transmitted horizontally to bystander cells, where they induce ongoing NM prion replication (Hofmann et al., 2013; Liu et al., 2016). While mammalian protein aggregates with prion-like domains can induce aggregation of homotypic proteins in recipient cells, it is unclear if such aggregates also begin propagating (again testable by monitoring maintenance of protein aggregates by progeny).

Of note, generation of cell clones continuously producing NM aggregates required production of stable cell lines, limiting dilution cloning, aggregate induction and a second round of cloning and thus took at least 3-4 months before experiments could be started. Similar experiments with mammalian proteins therefore require the establishment of comparable cellular models with mammalian proteins yet to be identified with true prion-like characteristics, a task that is beyond the scope of this study.

Interestingly, our NM interactomes also partially overlap with recently identified interactors of TDP-43 and the aggregating prion-like domain of TDP-43 (TDP-43 CTF) in the natural cell environment using proximity-dependent biotin identification (Chou et al., 2018). Of note, as RNA-binding proteins are part of submicroscopic protein complexes that would lead to comparable biotinylation results for soluble and aggregated TDP-43 species (Youn et al., 2018), co-localization by confocal microscopy was included for validation, just as in our study. Chou et al. showed recruitment of components of the nucleocytoplasmic transport (Nups), factors that were also recruited by our soluble (2 Nups, see suppl. Table) and aggregated NM-HA (6 Nups). Nups are also components of SGs (Jain et al., 2016; Markmiller et al., 2018). We have now included this information in the discussion section (p. 17).

Response to Reviewer #1:

Many proteins in the human proteome contain so-called Prion-like Domains (PrID): low complexity domains responsible for the formation of membrane-less organelles, such as stress granules (SGs). It has been proposed that dysregulation of such SGs can result in pathogenic protein aggregation, however, the molecular mechanisms mediating this process remains elusive. Here, the authors use the NM prion domain of the yeast Sup35 prion protein, which shows a remarkably similar amino acid composition to PrID-containing proteins, as a model protein to study this mechanism in a mammalian background.

Response: Of note, the Sup35 prion domain used here constitutes one of the few prion domains found in S. cerevisiae prions used to develop and train present prion algorithms (Alberti et al., 2009; King et al., 2012; Toombs et al., 2010). The definition of prion-like domains in mammalian proteins is therefor based on their compositional similarity with Sup35 and not vice versa. These yeast prion algorithms have been used to identify comparable domains in mammalian proteins (King et al., 2012). While mammalian proteins with PrLDs can undergo phase transition, it is unclear if any of these proteins behave as bona fide prions in mammals. Consequently, studying the prototype S. cerevisiae Sup35 prion domain for its prion behavior in the mammalian environment is vital to prove if prion algorithms can predict prion behavior in the mammalian context. Our study of interaction partners can help to unravel the cellular machinery that aids in replication of protein aggregates with prion-like domains in the mammalian cell environment.

They found that the NM prion (NM-HAagg) as well as the soluble NM (NM-HAsol) interactome are enriched for intrinsically disordered proteins, RNA-binding proteins and RNA, a highly similar composition as observed for SGs. Importantly, in contrast to the previously proposed mechanism of PrLD-containing proteins, where aberrant SG formation upon certain stress conditions can lead to formation of pathogenic inclusion, the authors here conclude that these proteins can be recruited to inclusions independent of SG formation.

1. Fig 4A shows the percentage of identified SG proteins in both groups (NM-HAsol and NM-HAagg) is roughly the same, however the identity of these proteins is not mentioned here. For clarity and to support the suggested hypothesis, the authors should include a Venn diagram of all SG proteins identified in both NM-HAsol and NM-HAagg interactomes. If the hypothesis is correct, and the recruitment of SG proteins in aggregates is a result of the interaction of these proteins with soluble NM, most proteins should be present in both groups.

Response: We have now included the suggested Venn diagram demonstrating that the majority of identified SG proteins interacts with both soluble and aggregated NM-HA (revised Figure 4B). Identified proteins are also shown in the supplementary excel table.

2. Upon oxidative stress, soluble NM was recruited into SGs (Fig 5D). The authors should show that this is a reversible process to exclude the possibility that NM aggregation occurs as a result of this stress condition.

Response: This is indeed an interesting question. For the present study, we have now included additional supplementary Fig. S4, demonstrating that recruitment of soluble NM to SGs in N2a cells is transient. Please note that we had to decrease the incubation time with arsenite to 30 min, as the 1 h treatment was toxic to N2a cells when cultured extended periods post arsenite treatment. This is a standard time frame for sublethal SG induction by arsenite (Kedersha et al., 2000) and leads to strong SG induction with less toxicity post treatment in our N2a cells. Further, we now also included arsenite treatment of N2a cells stably expressing soluble NM-GFP, to include a cell line expressing NM

with another tag. In both cases, NM was recruited to transient SGs that disassembled after stress subsided, with no persistent SGs remaining.

3. The SG proteins TIA-1 and TIAR, which clearly bind NM-HAagg in the in vitro pull-down and immunostaining assays, are not detected by the LC-MS/MS approach. The authors should elaborate on why they used these two proteins to perform these control experiments, instead of other SG components that are identified by the LC-MS/MS approach.

Response: Aim of our study was to demonstrate that NM in its soluble and aggregated state interacts with SG components and that soluble NM is recruited to SGs. For immunofluorescence and pull-downs, we therefore used canonical SG markers TIA1 and TIAR and identified RNA, which is a major component of SGs. We had, however, also included G3BP in pull-downs, which was found to interact with aggregated NM-HA by mass spectrometry (Figure 5B). We now demonstrate this interaction also using NM-GFP expressing cells (supplementary Fig. S3B-D).

Response to Reviewer #2 (Comments to the Authors (Required)):

The interesting article by Riemschoss and co-workers studies the effect of heterologous expression in cultured mouse cells of a Q/N-rich prion domain (NM) from yeast Sup35. Using cell biology (immuno-fluorescence) and systems (proteomics: affinity tag purification followed by mass spectrometry) approaches, the authors compare the portion of the mouse proteome that co-purifies with the yeast sequences, both in its soluble or aggregated states. The proteins identified were then compared with those characteristics of membrane-less stress (P)-granules. Transfection of in vitro-assembled NM fibrils, which templates the assembly of prion aggregates in the recipient cells forming two distinct lineages (in terms morphology and toxicity), does not result in induction of stress granules.

The article, although merely descriptive in a good part, is interesting. It builds on previous results by the same group reporting the successful propagation of the same yeast prion (Sup35) in mammalian cells, by contributing now the clue that amyloid aggregation does not necessarily imply the assembly of stress granules: the same proteins (many of them RNA binding proteins, such as TDP-43 and FUS, also carrying prion-like domains) that usually form these membrane-less organelles were identified in their analysis as sequestered at the prion-like aggregates. These aggregates also include proteins implicated in processes of protein quality control (proteostasis), tagging aggregates for autophagy and degradation at the proteasome.

The experimental evidence provided is technically sound and the manuscript deserves publication. However, the only doubt is whether the conclusions extracted from such heterologous studies can be fully translated into the natural, homotypic mammalian system. As the authors state at the Discussion section, the metazoan Hsp110/70/40 machinery that disassembles protein aggregates in the cytosol is unable to do so on those formed by Sup35 (NM). In addition, the same authors had shown that this yeast prion is not cytotoxic when expressed in mammalian cells. The answer should come from

complementary experiments not transfecting yeast prion aggregates, but aggregates from a disease-relevant mammalian protein expressed in the recipient cells.

Response: Aim of our study was to identify interactors of protein aggregates with prion-like domains that can adapt prion conformations in mammalian cells. Our previous studies have shown that Sup35 NM expressed in the cytosol of mammalian cells truly behaves as a prion, as it very rarely spontaneously aggregates, is inducible to form prions by recombinant NM fibrils, and faithfully propagates the heritable prion state to progeny and bystander cells (Hofmann et al., 2013; Krammer et al., 2009; Liu et al., 2016). NM aggregates are also non-toxic (Hofmann et al., 2013). These characteristics are hallmarks also of mammalian PrP-derived prions replicating in mitotically active N2a cells (Ghaemmaghami et al., 2007). We are unaware of studies demonstrating the same prion behavior for mammalian proteins with proposed prion-like domains. Thus, our model system, albeit artificial, represents an extremely useful model to study prion behavior of proteins with prion-like domains in the mammalian context.

Propagation of prions in yeast relies on fragmentation of prion fibrils by cellular disaggregase Hsp104- this factor is not present in the mammalian cytosol. While our study provides strong evidence that several components of the cellular protein quality control associate with NM prions, identification of specific factors or even of an interplay of these factors crucial for prion propagation is beyond the scope of this study. We agree with the reviewer that identifying such disaggregating factors is extremely interesting and this process is being addressed in our ongoing research. As we neither know the factors required to produce infectious cytosolic prions in mammals nor do we have a suitable cellular model to propagate self-templating protein aggregates composed of mammalian proteins with prion-like domains, future research is required to produce such data and models.

Response to Reviewer #3 (Comments to the Authors (Required)):

In the manuscript entitled "Fibril-induced glutamine-/ asparagine-rich prions sequester components of stress granules in mammalian cells" Riemschoss and colleagues studied the interaction partners of an HA-tagged NM domain (NM-HA) of the yeast prion Sup35 in mouse neuroblastoma (N2a) cells. They employed extracts prepared from stable cell lines containing either soluble or aggregated NM for an immunoprecipitation with anti-HA antibodies. The proteins present in the immunopellet were then identified by LC-MS/MS. This analysis revealed that the interactomes of soluble and 'prionized' NM-HA overlap with that of stress granules.

In my opinion the major shortcoming of the manuscript is that it is very descriptive with very little mechanistic insights. For example, one would like to see that the putative interaction of NM-HA with any of the identified proteins has functional consequences. In addition, it is difficult to appreciate that the analysis of the interactome of a yeast protein in mammalian cells will help to enhance our knowledge about physiological or pathophysiological activities of mammalian prion or prion-like proteins.

*Response: We agree with the reviewer that our model system is artificial. However, approx. 1% of mammalian proteins contain putative prion-like domains identified based on compositional similarity with Sup35 and few other yeast prions (Alberti et al., 2009; Toombs et al., 2010). Interest in this group of proteins is tremendous (213 Pubmed citations for a review on prion-like domains in mammalian proteins (King et al., 2012)). While some of these proteins form aggregates and can be induced to aggregate upon exposure of cells to aggregated homotypic proteins (for example, (Nonaka et al., 2013; Smethurst et al., 2016)), to the best of our knowledge for none of the proteins, prion behavior similar to mammalian or yeast prions has been documented. This would include: normally soluble state, rare induction of the prion state, **heritable** prion state in mitotically active cellular models, induction of **ongoing** aggregation in neighboring cells via cell contact or extracellular vesicles. If and how mammalian proteins with prion-like domains could adapt a true prion-like state is so far unknown. It is therefore important to understand general cellular mechanisms that allow propagation of proteins with prion-like domains in the mammalian context and to identify factors that interact with these protein aggregates and promote their prion behavior. For example, it might be possible that prions with a prion-like domain go unnoticed by the cellular quality control, which we show is not the case. Further, our data demonstrate that external seeds of homotypic proteins with prion-like domains can induce aggregates that share a SG interactome, thus demonstrating that sequestration of SG components by aggregates can be completely independent of SG assembly. This finding has general consequences for disease-associated protein aggregates for which misfolding due to recruitment to dysregulated SGs has been postulated (Li et al., 2013).*

- Throughout the manuscript the authors discuss their findings in the context of the mammalian prion protein (PrP) or mammalian RNA-binding proteins with low complexity prion-like domains. What is the advantage of an artificial system - the interactome of a yeast protein in mammalian cells - instead of analyzing directly interactors of either PrP or prion-like domains of the respective RNA-binding proteins?

*Response: Mammalian proteins with prion-like domains sharing compositional similarity to *S. cerevisiae* prion domains are abundant (“Prld” by prediction by prion algorithms (King et al., 2012)). Algorithms used to predict prion domains in mammalian proteins are based on very few yeast prion domains, including the archetypal Sup35 prion domain (Alberti et al., 2009; King et al., 2012; Toombs et al., 2010). Yeast prion domains do not share sequence similarity with mammalian PrP, except for the presence of repeats in some yeast prion domains. Replication of PrP^{Sc} prions occurs on the cell surface or along the endocytic pathway and thus differs from the proteins of interest here. We have discussed this in the introduction. We used the Sup35 NM domain as model protein to study potential prion behavior of such a domain in mammalian cells. To the best of our knowledge, this is the only prion-like domain shown so far that can undergo a full prion life cycle in mammalian cells and where cell populations are available that transmit aggregates to progeny (see above). For example, while TDP-43 can be induced to aggregate upon exposure to patient brain material or recombinant TDP-43 fibrils, these*

protein aggregates are toxic and their propagation (revealed by heritability in mitotically active cells) has not been demonstrated. The advantages of studying interactors of the Sup35 prion domain in mammalian cells are a) model protein with true prion behavior (Hofmann et al., 2013; Krammer et al., 2009; Liu et al., 2016) b) no loss-of-function phenotype (only compositional similarity to mammalian Prlds), c) no toxicity of aggregates, allowing generation of cell clones replicating NM prions, d) proof-of-principle that the prototype prion domain used for the development of prion algorithms indeed exhibits prion behavior in mammalian cells.

- The authors compare the immunopellets of a protein in two different states, soluble and aggregated. There are some technical problems with such an approach:

1. They have a control for unspecific interactions of the antibodies/beads in extracts containing the soluble NM-HA, however an appropriate control is missing for the extracts containing aggregated/'prionized' NM-HA. Thus, it remains to be proven that proteins identified in the N2a-NM-HAagg cells extracts are indeed specific interactors of the aggregated NM-HA.

Response: We agree with the reviewer. We have now included a control IP using unspecific IgG beads and N2a NM-HA^{agg} cell lysate to demonstrate that beads did not non-specifically pull out identified interactors (revised Fig. 6A). Further, we have included IgG controls for pull-downs of G3BP and p62 in lysates of cells producing aggregated NM-GFP (supplementary Fig. S3C).

2. The IP is critically dependent on the accessibility of the HA tag. As a consequence only a small subset of aggregated NM-HA with an exposed HA tag, maybe with very distinct and not representative properties, is immunoprecipitated.

Response: We have now performed control experiments using N2a cells stably expressing soluble NM-GFP and a cell clone persistently producing NM-GFP^{agg} (Hofmann et al., 2013). We show that also NM-GFP interacts with G3BP and p62 (supplementary Fig. S3).

3. N2a-NM-HAagg cells contain in addition to aggregated NM-HA also soluble NM-HA. This is the fraction after translation and before recruitment to NM-HA aggregates. Therefore, one has to consider that at least some of 'common' interactors identified N2a-NM-HAagg cells may have not interacted with aggregated NM-HA.

Response: We agree with the reviewer that there is a chance that some of the proteins we have identified in the lysates of cells with NM aggregates could be due to binding to the NM fraction which is not aggregated. This is a valid point and accounts for all studies on interactors of soluble and aggregated proteins (for example: TDP-43 (Chou et al., 2018); FUS (Kamelgarn et al., 2016); amyloidogenic model proteins (Olzscha et al., 2011)). Of note, SG components such as TIA-1, TIAR and G3BP are part of physiological submicroscopic RNA-protein granules that are present also in unstressed cells and can be sedimented already at 10.000 x g (Namkoong et al., 2018; Youn et al., 2018).

Differences in interaction partners are usually revealed by comparing soluble proteins and their aggregation-prone mutants or deletion mutants (Chou et al., 2018; Kamelgarn et al., 2016; Olzscha et al., 2011). In these cases, differences in the amino acid sequence of the compared proteins could also account for differences in interactors. Furthermore, in most studies, not all of the cells expressing aggregation-prone proteins produce aggregates. Our system has the advantage that the protein sequence is the same, and heritable protein aggregation was induced by exogenous seeds many cell divisions ago (Krammer et al., 2009). We used cell clones that produce NM aggregates over continuous culture for pull-downs for mass spec (Hofmann et al., 2013; Krammer et al., 2009; Liu et al., 2016). Cell clones had been selected because virtually all daughter cells produce aggregates even after years of culture (Hofmann et al., 2013). SDD-AGE demonstrated that the vast majority of NM-HA is present in the aggregated fraction (Fig. 1C). We have now included a sentence on page 8 to more clearly point this out (page 8, second sentence “While NM-HA^{sol}”.

Mass spec analysis identified SG components and proteins of the quality control machinery in all pull-downs. We had validated the interaction of aggregated NM with SG components (FUS, TDP-43 and TIA-1, RNA) by confocal microscopy. While we cannot exclude the possibility that some of our candidate interactors might only interact with soluble NM-HA (in agreement with other studies (Chou et al., 2018)), we have demonstrated that several SG markers can be found within the NM-HA aggregates. Thus, our conclusion that NM prions sequester SG components is valid. To clarify, we have added a sentence on page 11 stating that there is the possibility that some of the interactors identified in the NM-HA^{agg} sample might be actually interacting with NM-HA^{sol}.

References

- Alberti, S., Halfmann, R., King, O., Kapila, A., and Lindquist, S. (2009). A systematic survey identifies prions and illuminates sequence features of prionogenic proteins. *Cell* 137, 146-158.
- Chou, C.C., Zhang, Y., Umoh, M.E., Vaughan, S.W., Lorenzini, I., Liu, F., Sayegh, M., Donlin-Asp, P.G., Chen, Y.H., Duong, D.M., et al. (2018). TDP-43 pathology disrupts nuclear pore complexes and nucleocytoplasmic transport in ALS/FTD. *Nat Neurosci* 21, 228-239.
- Duernberger, Y., Liu, S., Riemschoss, K., Paulsen, L., Bester, R., Kuhn, P.H., Scholling, M., Lichtenthaler, S.F., and Vorberg, I. (2018). Prion Replication in the Mammalian Cytosol: Functional Regions within a Prion Domain Driving Induction, Propagation, and Inheritance. *Mol Cell Biol* 38.
- Ghaemmaghami, S., Phuan, P.W., Perkins, B., Ullman, J., May, B.C., Cohen, F.E., and Prusiner, S.B. (2007). Cell division modulates prion accumulation in cultured cells. *Proc Natl Acad Sci U S A* 104, 17971-17976.
- Hofmann, J.P., Denner, P., Nussbaum-Krammer, C., Kuhn, P.H., Suhre, M.H., Scheibel, T., Lichtenthaler, S.F., Schatzl, H.M., Bano, D., and Vorberg, I.M. (2013). Cell-to-cell propagation of infectious cytosolic protein aggregates. *Proc Natl Acad Sci U S A* 110, 5951-5956.
- Jain, S., Wheeler, J.R., Walters, R.W., Agrawal, A., Barsic, A., and Parker, R. (2016). ATPase-Modulated Stress Granules Contain a Diverse Proteome and Substructure. *Cell* 164, 487-498.
- Kamelgarn, M., Chen, J., Kuang, L., Arenas, A., Zhai, J., Zhu, H., and Gal, J. (2016). Proteomic analysis of FUS interacting proteins provides insights into FUS function and its role in ALS. *Biochim Biophys Acta* 1862, 2004-2014.
- Kedersha, N., Cho, M.R., Li, W., Yacono, P.W., Chen, S., Gilks, N., Golan, D.E., and Anderson, P. (2000). Dynamic shuttling of TIA-1 accompanies the recruitment of mRNA to mammalian stress granules. *J Cell Biol* 151, 1257-1268.
- King, O.D., Gitler, A.D., and Shorter, J. (2012). The tip of the iceberg: RNA-binding proteins with prion-like domains in neurodegenerative disease. *Brain Res* 1462, 61-80.
- Krammer, C., Kryndushkin, D., Suhre, M.H., Krammer, E., Hofmann, A., Pfeifer, A., Scheibel, T., Wickner, R.B., Schatzl, H.M., and Vorberg, I. (2009). The yeast Sup35NM domain propagates as a prion in mammalian cells. *Proc Natl Acad Sci U S A* 106, 462-467.
- Krauss, S., and Vorberg, I. (2013). Prions Ex Vivo: What Cell Culture Models Tell Us about Infectious Proteins. *Int J Cell Biol* 2013, 704546.
- Li, Y.R., King, O.D., Shorter, J., and Gitler, A.D. (2013). Stress granules as crucibles of ALS pathogenesis. *J Cell Biol* 201, 361-372.

Liu, S., Hossinger, A., Hofmann, J.P., Denner, P., and Vorberg, I.M. (2016). Horizontal Transmission of Cytosolic Sup35 Prions by Extracellular Vesicles. *MBio* 7.

Markmiller, S., Soltanich, S., Server, K.L., Mak, R., Jin, W., Fang, M.Y., Luo, E.C., Krach, F., Yang, D., Sen, A., *et al.* (2018). Context-Dependent and Disease-Specific Diversity in Protein Interactions within Stress Granules. *Cell* 172, 590-604 e513.

Namkoong, S., Ho, A., Woo, Y.M., Kwak, H., and Lee, J.H. (2018). Systematic Characterization of Stress-Induced RNA Granulation. *Mol Cell* 70, 175-187 e178.

Nonaka, T., Masuda-Suzukake, M., Arai, T., Hasegawa, Y., Akatsu, H., Obi, T., Yoshida, M., Murayama, S., Mann, D.M., Akiyama, H., *et al.* (2013). Prion-like properties of pathological TDP-43 aggregates from diseased brains. *Cell Rep* 4, 124-134.

Olzscha, H., Schermann, S.M., Woerner, A.C., Pinkert, S., Hecht, M.H., Tartaglia, G.G., Vendruscolo, M., Hayer-Hartl, M., Hartl, F.U., and Vabulas, R.M. (2011). Amyloid-like aggregates sequester numerous metastable proteins with essential cellular functions. *Cell* 144, 67-78.

Rujano, M.A., Bosveld, F., Salomons, F.A., Dijk, F., van Waarde, M.A., van der Want, J.J., de Vos, R.A., Brunt, E.R., Sibon, O.C., and Kampinga, H.H. (2006). Polarised asymmetric inheritance of accumulated protein damage in higher eukaryotes. *PLoS Biol* 4, e417.

Smethurst, P., Newcombe, J., Troakes, C., Simone, R., Chen, Y.R., Patani, R., and Sidle, K. (2016). In vitro prion-like behaviour of TDP-43 in ALS. *Neurobiol Dis* 96, 236-247.

Toombs, J.A., McCarty, B.R., and Ross, E.D. (2010). Compositional determinants of prion formation in yeast. *Mol Cell Biol* 30, 319-332.

Youn, J.Y., Dunham, W.H., Hong, S.J., Knight, J.D.R., Bashkurov, M., Chen, G.I., Bagci, H., Rathod, B., MacLeod, G., Eng, S.W.M., *et al.* (2018). High-Density Proximity Mapping Reveals the Subcellular Organization of mRNA-Associated Granules and Bodies. *Mol Cell* 69, 517-532 e511.

June 21, 2019

RE: Life Science Alliance Manuscript #LSA-2018-00280R

Dr. Ina Vorberg
German Center for Neurodegenerative Diseases (DZNE)
Cell Biology and Pathophysiology of Prions
Sigmund-Freud-Strasse 27
Bonn 53127
Germany

Dear Dr. Vorberg,

Thank you for submitting your revised manuscript entitled "Fibril-induced glutamine-/asparagine-rich prions recruit stress granule proteins in mammalian cells". As you will see, the reviewers appreciate the introduced changes and we would thus be happy to publish your paper in Life Science Alliance pending final revisions necessary to meet our guidelines:

- I would like to request from you source data for the IPs in Fig 5B and S2A
- please add a callout in the text to figure S5

A. FINAL FILES:

-- Summary blurb (enter in submission system): A short text summarizing in a single sentence the study (max. 200 characters including spaces). This text is used in conjunction with the titles of papers, hence should be informative and complementary to the title. It should describe the context and significance of the findings for a general readership; it should be written in the present tense

and refer to the work in the third person. Author names should not be mentioned.

B. MANUSCRIPT ORGANIZATION AND FORMATTING:

Sincerely,

Reviewer #1 (Comments to the Authors (Required)):

The authors provide a satisfactory answer to my three main remarks and included additional data to support these answers. Although I agree with reviewers #2 and #3 that this is a merely descriptive manuscript, it does provide some interesting insights and I do believe it fits into the scope of Life Science Alliance. Therefore, I believe the manuscript is now sufficiently developed for publication in Life Science Alliance.

Reviewer #2 (Comments to the Authors (Required)):

The authors have addressed all the points I had raised in my previous review.

Reviewer #3 (Comments to the Authors (Required)):

most of my concerns from my previous review have been adequately addressed

June 25, 2019

RE: Life Science Alliance Manuscript #LSA-2018-00280RR

Dr. Ina Vorberg
German Center for Neurodegenerative Diseases (DZNE)
Cell Biology and Pathophysiology of Prions
Sigmund-Freud-Strasse 27
Bonn 53127
Germany

Dear Dr. Vorberg,

Thank you for submitting your Research Article entitled "Fibril-induced glutamine-/asparagine-rich prions recruit stress granule proteins in mammalian cells". It is a pleasure to let you know that your manuscript is now accepted for publication in Life Science Alliance. Congratulations on this interesting work.

DISTRIBUTION OF MATERIALS:

Again, congratulations on a very nice paper. I hope you found the review process to be constructive and are pleased with how the manuscript was handled editorially. We look forward to future exciting submissions from your lab.

Sincerely,

Andrea Leibfried, PhD
Executive Editor
Life Science Alliance
Meyrhofstr. 1
69117 Heidelberg, Germany
t +49 6221 8891 502
e a.leibfried@life-science-alliance.org
www.life-science-alliance.org